# *Bacillus licheniformis*: A Producer of Antimicrobial Substances, including Antimycobacterials, Which Are Feasible for Medical Applications

**DOI:** 10.3390/pharmaceutics15071893

**Published:** 2023-07-05

**Authors:** Margarita O. Shleeva, Daria A. Kondratieva, Arseny S. Kaprelyants

**Affiliations:** A.N. Bach Institute of Biochemistry, Federal Research Centre ‘Fundamentals of Biotechnology’, Russian Academy of Sciences, 119071 Moscow, Russia

**Keywords:** *Bacillus licheniformis*, *Mycobacterium tuberculosis*, bacteriocin, antimicrobial peptides

## Abstract

*Bacillus licheniformis* produces several classes of antimicrobial substances, including bacteriocins, which are peptides or proteins with different structural composition and molecular mass: ribosomally synthesized by bacteria (1.4–20 kDa), non-ribosomally synthesized peptides and cyclic lipopeptides (0.8–42 kDa) and exopolysaccharides (>1000 kDa). Different bacteriocins act against Gram-positive or Gram-negative bacteria, fungal pathogens and amoeba cells. The main mechanisms of bacteriocin lytic activity include interaction of peptides with membranes of target cells resulting in structural alterations, pore-forming, and inhibition of cell wall biosynthesis. DNase and RNase activity for some bacteriocines are also postulated. Non-ribosomal peptides are synthesized by special non-ribosomal multimodular peptide synthetases and contain unnatural amino acids or fatty acids. Their harmful effect is due to their ability to form pores in biological membranes, destabilize lipid packaging, and disrupt the peptidoglycan layer. Lipopeptides, as biosurfactants, are able to destroy bacterial biofilms. Secreted polysaccharides are high molecular weight compounds, composed of repeated units of sugar moieties attached to a carrier lipid. Their antagonistic action was revealed in relation to bacteria, viruses, and fungi. Exopolysaccharides also inhibit the formation of biofilms by pathogenic bacteria and prevent their colonization on various surfaces. However, mechanism of the harmful effect for many secreted antibacterial substances remains unknown. The antimicrobial activity for most substances has been studied in vitro only, but some substances have been characterized in vivo and they have found practical applications in medicine and veterinary. The cyclic lipopeptides that have surfactant properties are used in some industries. In this review, special attention is paid to the antimycobacterials produced by *B. licheniformis* as a possible approach to combat multidrug-resistant and latent tuberculosis. In particular, licheniformins and bacitracins have shown strong antimycobacterial activity. However, the medical application of some antibacterials with promising in vitro antimycobacterial activity has been limited by their toxicity to animals and humans. As such, similar to the enhancement in the antimycobacterial activity of natural bacteriocins achieved using genetic engineering, the reduction in toxicity using the same approach appears feasible. The unique capability of *B. licheniformis* to synthesize and produce a range of different antibacterial compounds means that this organism can act as a natural universal vehicle for antibiotic substances in the form of probiotic cultures and strains to combat various types of pathogens, including mycobacteria.

## 1. Introduction

The spread of bacterial strains that cause severe infectious diseases but are now resistant to known antibiotics necessitates the search for and development of new approaches to combat these diseases [1]. The most known and medically important example that illustrates this problem is the growing number of cases of multidrug-resistant strains of *Mycobacterium tuberculosis* (*Mtb*), which is the causative agent of tuberculosis. In addition to drug resistance, *Mtb* is able to asymptomatically persist in the host organism for many years, causing latent forms of tuberculosis. In this dormant state, *Mtb* cells are also resistant to known antibiotics [2,3,4].

The search for and study of substances that have bactericidal or bacteriostatic properties against human and animal pathogens are also required for the development of new antibiotic therapy or disinfectants for objects and surfaces that have been in close contact with patients and therefore may carry pathogenic bacteria. Currently, in addition to the synthesis of new chemical substances, considerable attention has focused on the analysis of the potential of natural products from different origins as antimicrobials. The discovery of antibiotics with activity against human pathogens is often based on the observation of the interaction between microorganisms, called antagonism. This antagonism manifests through the synthesis and release of substances that inhibit or completely suppress the growth of other species. Under natural conditions, a microorganism-secreted substance(s) that inhibits the growth of another organism gains a competitive advantage in the struggle for environmental resources. Most of the antibiotics used for medical applications are secreted products or derivatives of microorganisms belonging to the order *Actinomycetalis* (among them, the most well-known are *Streptomyces*). The bacterial world represents a huge reservoir of not-yet-discovered and used substances that have antibacterial potential. In this regard, representatives of the genus *Bacillus* are known as producers of many enzymes and antimicrobial compounds. For example, *Bacillus amyloliquefaciens* is a source of the natural antibiotic barnase (ribonuclease), alpha-amylase, which is used in starch hydrolysis; protease subtilisin, which is used in combination with detergents; and the restriction enzyme, BamH1, used in DNA research [5]. *Bacillus subtilis* produces 66 derived antimicrobials, and *Bacillus brevis* produces 23 peptide antibiotics [6]. There is a growing interest in considering these substances, including bacteriocins, as alternative antimicrobials for the treatment of human and animal infections [7,8,9,10,11].

Currently, the use of bacterial probiotic strains and their metabolic products is considered a new approach for the control and prevention of various infectious diseases [12]. Animal studies have demonstrated that probiotics from the *Bacillus* genus have antimicrobial properties. This conclusion also applies to humans [13,14]. The use of bacteriocins and antimicrobial peptides produced by probiotic strains is a suitable alternative to antibiotics because their production is inexpensive and resistance to them is rare [15]. They exhibit a broad spectrum of activity against many Gram-positive and Gram-negative bacteria and fungi. Owing to the efficacy and cost-effectiveness of many of these compounds, they are attractive for clinical use [16]. A few natural peptides have shown potential because of their desirable therapeutic properties, including antimicrobial, antiviral, anticancer, and contraceptive activities. Additionally, they have been shown to protect against topical and systemic infections in combination with conventional antibiotics [17].

Among the organisms belonging to the *Bacillus* genus, *Bacillus licheniformis* is a unique specie that which produces wide variety of antimicrobial substances. This bacterium shows promise for use as a probiotic in the treatment of dysbacteriosis, which is caused by various diseases [13]. The effectiveness of *B. licheniformis* as a probiotic is associated with its ability to produce a large amount of substances with antimicrobial, antioxidant and immunomodulatory activities [13], for example, a phosphorus-containing triene antibiotic called proticin [18,19]. *B. licheniformis* shows a protective effect in zebrafish (Danio rerio) against *Vibrio parahaemolyticus* infection. Due to the antagonistic activity of this probiotic, the complete survival of infected fish was observed in contrast with untreated fish [20]. This probiotic, in combination with *Bifidobacterium breve*, significantly inhibited the adhesion of the pathogen *Kocuria rhizophila* in vitro [21] and showed antivibrio activity against *Vibrio parahaemolyticus* [22]. The use of a crude extract from *B. licheniformis* resulted in marked antiviral activity against porcine epidemic diarrhea virus in Vero cells and reduced virus shedding in piglets [23]. After the administration of fermented *B. licheniformis* products, the number of pathogenic bacteria including *Clostridium perfringens* significantly decreased in cats with chronic diarrhea [24]. In piglets, *B. licheniformis* treatment showed positive effects against *Salmonella* [25]. Probiotic *B. licheniformis* produces antimicrobial substances and has a strong ability to auto- and coaggregate against pathogenic bacteria [26]. Approaches are being developed to combat bacterial biofilms using silver nanoparticles and the probiotic *B. licheniformis* [27].

The bacteriocins from *B. licheniformis* are being considered as natural preparations as a preservative in the food industry [28,29].

In general, bacteriocins are a group of antimicrobial peptides that represent a potential alternative to classical antibiotics in the fight against antimicrobial resistance in pathogenic microorganisms. Several reports have been published in the literature about numerous bacteriocins, many of which currently remain undiscovered due to the wide variety of their natural sources; hence, further research in this area is required [11].

Considering the medical and industrial application of *Bacillus licheniformis*, a thorough description and characterization of the variety of antimicrobial compounds produced and their use against resistant pathogens, such as mycobacteria, are required.

Therefore, this review focused on the current state of knowledge about the classes of antibiotic substances produced by *B. licheniformis* and their structure and properties, which may allow a more comprehensive perspective of their antimicrobial substances, including their antimycobacterial properties.

## 2. Antibacterial Substances Secreted by *Bacillus licheniformis*

The endospore-forming bacterium *Bacillus licheniformis* is capable of producing a large amount of substances with different structures and different antibacterial activities [30] (Figure 1). When grown on identical media, different strains of *B. licheniformis* produce different sets of substances with antibacterial activities [31]. It is plausible that all strains of *B. licheniformis* potentially are able to produce variety of antimicrobial substances, however, the synthesis and production of particular substance can be differently regulated on the transcriptional or translational level for certain strains grown in identical media. As a result, the amount of secreted antimicrobial compounds may substantially vary from strain to strain allowing to consider them as unique producers of defined set of antimicrobials. The secreted antimicrobial substances have molecular masses ranging from 1.4 to 20 kDa [28,29,32,33,34,35,36,37,38].

Changing the composition of the medium used for *B. licheniformis* growth results in the alterations in the secreted substances. Thus, on media containing iron, *B. licheniformis* can synthesize the red pigment pulcherrimin [39]. When grown on a medium with lactate and a high ratio of nitrogen and carbon, *B. licheniformis* strain N.C.T.C. 7072 produces licheniformins; when grown on a medium with glucose and a low nitrogen/carbon ratio, this strain produces bacitracins [37]. Several substances synthesized by *B. licheniformis* have been investigated as antibiotics against various types of bacteria. They are listed and characterized below. Some of them (bacitracin) are used in combined antibacterial preparations intended for topical use. Others are used as oral antibiotics, but only in animals due to their toxic effects.

Among the antimicrobial components (Figure 1) produced by various strains of *B. licheniformis* in a nutrient medium, several groups differ in properties and structure.

### 2.1. Bacteriocins

Bacteriocins are substances represented by an amino acid sequence (peptides or proteins) that act against other strains of bacteria or closely related species. They demonstrate both bactericidal and bacteriostatic effects. Bacteriocins are natural antimicrobial peptides that are ribosomally synthesized by bacteria [10,11,40]. Genes whose expression leads to the synthesis of bacteriocins are organized into clusters of operons and can be located in the genome, plasmids, or other mobile genetic elements. These genes are inducible; peptide secretion and accumulation outside the cell are required for their induction. More details regarding bacteriocins biosynthesis are described in the review by Nishie et al. [9]. Bacteriocins are heterogeneous substances that demonstrate various biochemical properties, molecular weights, inhibitory spectra and mechanisms of action [10,41]. Due to the wide spectrum of antagonistic activity inherent to the bacteriocins of some strains of microorganisms, they have the potential to be used as a component of antibacterial drugs. The resistance to enzyme activity of the antimicrobial peptides produced by *Bacillus* spp. differs with stability over a wide range of pH and temperature. Most of these peptides have high specificity against microbial pathogens and low cytotoxicity against human cells [42]. The related bactericidal mechanisms include the pore-forming type, nuclease type with DNase and RNase functions, and the peptidoglycanase type [10] (Figure 2).

As a result, pores are formed, which leads to the rapid removal of small cytoplasmic molecules, ions from target cells, and the collapse of the protonmotive force, leading to the death of bacterial cells [9,43]. However, other antimicrobial mechanisms of bacteriocins were also identified [11]. Despite the recent popularity of research on the properties of bacteriocins and their use in medicine, veterinary sciences, and food industry [10,11], many bacteriocins have not yet been studied.

The production of several bacteriocin-like substances (Table 1) with different characteristics and a wide spectrum of activity against pathogenic bacteria was recorded in the strains of *B. licheniformis* [44]. For example, *B. licheniformis* SMIA-2, a thermophilic and thermostable enzyme-producing strain, is active against some strains of *Staphylococcus aureus* and *Bacillus* sp. The genome annotation of this strain detected gene clusters responsible for antimicrobial component production (lichenysin, fengycin, lichenicidin and bacillibactin biosynthetic gene clusters) [45].

*B. licheniformis* produces various bacteriocins ranging in molecular weight from 1.4 to 55 kDa, but the expression of a particular antimicrobial agent may depend on environmental conditions, growth period, and the specific bacterial strain [28,32,33,36,46].

In general, based on their thermostability, size and chemical moieties, bacteriocins are classified into four major groups [47]: class I, heat-stable lanthionine-containing peptides smaller than 5 kDa; class II, heat-stable non-lanthionine peptides smaller than 10 kDa; class III, heat-labile proteins larger than 30 kDa; and class IV, complex with a single lipid or carbohydrate moiety [48]. In this review, to describe the antimicrobial substances produced by different strains of *B. licheniformis*, Cotter’s classification is used, with a slight modification: heat-stable and heat-labile proteins larger than 10 kDa are assigned to class III, and class V was added, which includes proteins with undetermined molecular weight (Table 1, Figure 1).
pharmaceutics-15-01893-t001_Table 1Table 1Antimicrobial bacteriocins produced by *Bacillus licheniformis*.2.1.1. Class I: Heat-Stable Lanthionine-Containing Peptides Smaller Than 5 kDASubstance(s) Specific/Unspecific NameProducing StrainMolecular MassActivity AssayReferenceSublichenin*B. licheniformis* MCC 25123.348 kDa*Kocuria rhizophila* ATCC 9341*Pediococcus lolii* MCC 2972 *Enterococcus durans* B20G1*Enterococcus faecalis* MF3 *E. faecalis* MM2 *E. faecalis* CHL1 *E. faecalis* CHL3 *E. faecalis* CHL*E. faecalis* MCC 3063 *E. faecalis* MCC 2773*Enterococcus faecium* MCC 2763 *Entercoccus avium* CS32*Enterococcus cecorum* 1-40a *Lactobacillus plantarum* MCC 2774*Listeria monocytogenes*
*Staphylococcus aureus**Staphylococcus aureus* (MRSA)[49]*Escherichia coli**Klebsiella pneumoniae*Lichenicidin*Bacillus licheniformis* DSM 13(also produced by ATCC 14580, VK21,WIT 562, 564 and 566 strains, IMF20, IMF66, IMF69 and IMF80, I89)3 kDa and 3.25 kDa*Bacillus cereus* DSM 31*Bacillus halodurans* DSM 18197*Bacillus megaterium KM* (ATCC 13632)*Bacillus subtilis 168* (DSM 402)*Bacillus spec.* HIL Y-85,54728 *Enterococcus faecium* BM 4147–1*Enterococcus faecium* L4001*Lactobacillus sake* 790 E2 *Lactococcus lactis* NCTC 497 *Micrococcus luteus* DSM 1790 *Micrococcus luteus* ATCC 4698 *Staphylococcus aureus* ATCC 33592 (MRSA)*S. aureus ATCC* 29213 (MSSA) *S. aureus* 1450/94*S. aureus* Cowan (ATCC 12598) *S. aureus* Newman (NCTC 8178) *S. aureus* SG511*S. aureus Wood 46* (ATCC 10832)*Staphylococcus carnosus* TM300 *Staphylococcus gallinarum* Tü 928 *Staphylococcus saprophyticus* DSM 20229*Staphylococcus simulans* 22 *S. aureus LT440/09* (community acquired MRSA)*S. aureus* LT420/09 (MRSA)*S. aureus LT819/09* (MRSA, Rhine-Hessen epidemic strain)*Enterococcus faecalis**Streptococcus agalactiae**B. subtilis* L1*Rhodococcus* sp. strain SS2 *M. luteus* B1314*B. megaterium* VKM41 *B. pumilus* 2001 *B. globigii* I*B. amyloliquefaciens* I*M. smegmatis* 1171*M. phlei* 1291[50,51,52,53,54,55]**2.1.2. Class II: heat-stable non-lanthionine peptides smaller than 10 kDa*****2.1.2.1. Peptides active only against Gram-positive microorganisms***Bacillocin 490*B. licheniformis* 490/52 kDa*Bacillus licheniformis* 5 A2*Listeria innocua*
*Staphylococcus epidermidis*
*Bacillus anthracis* 7700*Bacillus subtilis* AZ56*Bacillus cereus* 6A2*Bacillus stearothermophilus* 9A19 *Bacillus smithii* PRO/S[28]Bacteriocin-like substance*Bacillus licheniformis* H13.5 kDa*E. faecalis* ATCC 19433*L. monocytogenes* ATCC 19111*B. cereus* ATCC 14579 *B. subtilis* ATCC 6633*Lactobacillus species* ATCC 33198*Lactobacillus fermentum*[56]*P. fluorescens*Bacteriocin-like antibacterial peptides*B. licheniformis* AnBa9<10 kDa*Staphylococcus aureus* GCS1*Bacillus cereus* GCS2*Staphylococcus epidermidis* GCS4*Kurthia gibsonii* GCS6*Micrococcus luteus* GCS7*Streptococcus mitis* GCS9*Bacillus subtilis* B-4219*L. lactis* B-1821*Staphylococcus epidermidis* B-4268*Bacilus smithii* NRS-173*Lactobacillus acidophilus* B-4495 *Micrococcus luteus* B-287*Pediococcus acidilactici* B-14958 *Leuconostoc mesenteriodes*[35]Lichenin*B. licheniformis* 26L10/3RA1.4 kDa*Streptococcus bovis* SB3 *Streptococcus bovis* 26*Ruminococcus avefaciens* OF-2 *Ruminococcus avefaciens* C94 *Ruminococcus albus* B119 *Ruminococcus albus* A-6*Butyrivibrio fibrisolvens* OR 12*Eubacterium ruminantium* GA-195 *Lactobacillus casei* ED-108[32]Bacteriocin BL8*B. licheniformis* BTHT81.4 kDa*Clostridium perfringens**Staphylococcus aureus**Bacillus cereus**Bacillus circulans**Bacillus pumilus*[57]BSCY2*B.licheniformis* CY26.5 kDa*B.subtilis* 6633[58]Licheniocin50.2*B. licheniformis* VPS50.23.25 kDa*Bacillus subtilis* ATCC 6633*B. subtilis* 168*B. subtilis* W23*Enterococcus faecalis* ATCC 29212*Enterococcus saccharolyticus* ATCC 43076*Lactobacillus plantarum* LMG92088 *Lactobacillus zeae*
*Lactococcus lactis* IL1403 *Listeria monocytogenes* ATCC 19111 *Micrococcus luteus* ATCC 7468*Staphylococcus aureus* ATCC 25923 *Staph. aureus* ATCC 33591 *Streptococcus agalactiae* ATCC 12386[59]***2.1.2.2. Peptides active against both Gram-positive and Gram-negative microorganisms***Bacteriocin like inhibitory substance (BLIS)*Bacillus licheniformis* IITRHR2 (FJ447354)1.2 kDa*Bacillus cereus* MTCC 1305 *Bacillus subtilis* MTCC 736*Bifidobacterium bifidum* NCDC 235*Enterococcus faecalis* MTCC 439*Enterococcus faecalis* NCDC1 14 *Lactobacillus casei* NCDC017 *Lactobacillus lactis* NCDC094*Leuconostoc mesenteroides* NCDC 219 *Listeria monocytogenes* MTCC 387*Listeria monocytogenes* MTCC1 143 *Pediococcus pentosaceus* NCDC 273*Staphylococcus thermophilus* NCDC 074[60]*Escherichia coli* MTCC 1687*Pseudomonas aeruginosa* MTCC 9027*Shigella flexneri* MTCC 1457*Shigella sonnei* MTCC 2957Bacteriocin MKU3*B. licheniformis* MKU31.5 kDa*Bacillus subtilis* B4219 *Bacillus smithii* NRS173*Lactobacillus acidophilus* B4495 *Lactobacillus fermentum* B1840*Lactobacillus lactis* B1821*Staphylococcus epidermidis* B4268 *Micrococcus luteus* B287*Leuconostoc mesenteriodes* B1118 *Pediococcus acidilactici* B14958*Staphylococcus aureus* GCS1 *Bacillus cereus* GCS2*Bacillus cereus* GCS3*Staphylococcus epidermidis* GCS4 *Staphylococcus epidermidis* GCS5*Kurthia gibsonii* GCS6*Micrococcus luteus* GCS7 *Bacillus subtilis* GCS8*Streptococcus fecalis* GCS9 *Bacillus cereus* GCS10*Bacillus cereus* GCS11*Lactobacillus acidophilus* GCS12[34]*Escherichia coli DH5a*Bacteriocin-like substance*B. licheniformis* B1164 kDa*B. cereus* CGMCC1.230*Listeria monocytogenes* CVCC1599*Micrococcus luteus* CMCC28001*S. aureus* CMCC26003*S. aureus* CICC21601*S. aureus* CVCC1885*Streptococcus equi* subsp. *zooepidemicus* CVCC1903[61]*E. coli* CVCC245*E. coli* CICC21525*E. coli* CVCC195*E. coli* CVCC249*S. enterica* ser. *Pullorum* CVCC79301*S. enterica* ser. *typhimurium* CVCC541dppABP*B. licheniformis* Me1 (MCC 2016)Between 3 and 3.5 kDa*L. innocua* FB 21 *L. murrayi* FB 69*M. luteus* ATCC 9341*L. monocytogenes* Scott A*Staph. aureus* FRI 722*B. cereus* F 4433*K. rhizophila* ATCC 9341[62,63,64]*Salm. typhimurium* MTCC 1251, FB 231*Salm. paratyphi* FB 254*E. coli* CFR 02 *Y. enterocolitica* MTCC 859*Shigella flexineri* (clinical isolate)Licheniformins A,B,C*B. licheniformis*NCTC 70723.8–4.8 kDa*Mycobacterium phlei**Staphylococcus aureus*[37,65]*E. coli*Antimicrobial compound*Bacillus licheniformis* MCC25146.5 kDa*M. luteus* ATCC9341*S. aureus* FRI722[31]*Klebsiella* sp.*A. hydrophila* NRRL B445**2.1.3. Class III: heat-stable and heat-labile proteins larger than 10 kDa**BLIS_SXAU06*B. licheniformis* SXAU0614 kDa*S. aureus**S. epidermidis**M. luteus**L. monocytogenes*[66]BL-DZ1 (BL00275)*B. licheniformis* strain D114 kDa*Bacillus pumilus* TiO1*Candida albicans* BH[67]*Pseudomonas aeruginosa PAO1 *(biofouling)YbdN Protein*B. licheniformis* (seaweed isolate)30.7 kDaMRSA 9551MRSA J2407VRE 788VRE 1349 *L. monocytogenes* NCTC 7973, NCTC 10357[68]Chitinase*B. licheniformis* MY75(also produced by Mb-2, TP–1, S213, SSCL-10, B307 strains)55 (67,68/62/60,65,66) kDa*G. saubinetii**A. niger*(*Phoma medicaginis*)[69,70,71,72,73,74,75]Antifungal Protein F2*B. licheniformis* BS-331 kDa*Aspergillus niger**Magnaporthe oryzae**Rhizoctonia solani**Fusarium oxysporum*[76]Antimicrobials protein*B. licheniformis JS*16 kDa*Bacillus cereus**Bacillus subtilis*[77]*Shigella dysenteriae**Salmonella typhimurium*AMS*B. licheniformis* T6-520 kDa*Desulfovibrio alaskensis* NCIMB 1349[36,78]AMS*B. licheniformis*H2O-1Between 90 and 120 kDa*Desulfovibrio alaskensis* NCIMB 1349SRB-containing consortium T6-lab[36,78]**2.1.4. Class IV: complex with a single lipid or carbohydrate moiety**F4, F5 and F6*B. licheniformis* BFP011Less than 45 kDa*B. amyloliquefaciens* TISTR 1045*B. licheniformis* TISTR 1010*B. subtilis* ATCC 6633*B. subtilis* TISTR 008*B. pumilus* TISTR 905*B. cereus* ATCC 11778*B. megaterium* (clinical isolate)*S. aureus* ATCC 25923[79]*E. coli* O157: H7*V. cholerae* (clinical isolate)*K. pneumonia* ATCC 17736*S. typhi* ATCC 5784Ieodoglucomides A and B*B. licheniformis* 09IDYM23ND*S. aureus**B. subtilis**B. cereus*[80]*E. coli**P. aeruginosa**S. typhi*Ieodoglucomide Candieodoglycolipid*B. licheniformis* 09IDYM23ND*Staphylococcus aureus**Bacillus subtilis**Bacillus cereus*[81]*Escherichia coli**Pseudomonas aeruginosa**Salmonella typhi***2.1.5. Class V: bacteriocins with undetermined molecular weight**Antipathogenic Metabolites*Bacillus licheniformis* (Upper arm skin isolate)ND*Staph. aureus* ATCC 6538[82]*Kl. Pneumoniae* subsp.*pneumonia* CMSOGHAntipathogenic Metabolites*Bacillus licheniformis* (Upper arm skin isolate)ND*Kl. Pneumoniae* subsp.*pneumoniae*[83]Antimicrobial substance*B. licheniformis* A-1-5B-APND*Prevotella intermedia* 1/P *Streptococcus mutans* ATCC 35668*Micrococcus luteus* DSM 1790[84]*Porphyromonas gulae* 3/HBacteriocin*B. licheniformis* HJ2020 MT192715.1ND*Staphylococcus aureus**Bacillus cereus**Bacillus subtilis*[85]*Escherichia coli* 0157:H7*Salmonella typhi**Pseudomonas aeruginosa*ND—no data. Gram- bacteria are below the line, and Gram+ bacteria are above the line. 


#### 2.1.1. Class I: Heat-Stable Lanthionine-Containing Peptides Smaller Than 5 kDa

Lantibiotics are antimicrobial peptides that undergo post-translational modification. They contain non-standard amino acids: lanthionine, β-methyl lanthionine and dehydrated residues (dehydrated amino acids) [86]. Their molecular weight does not exceed 5 kDa. Lantibiotics are active at low concentrations and are therefore attractive antimicrobials. They mainly target lipid II. A number of lantibiotics interact with the cell wall precursor lipid II (undecaprenyl-pyrophosphoryl-MurNAc-(pentapeptide)-GlcNAc), which prevents cell wall biosynthesis and contributes to the destruction of the bacterial membrane [87]. Thus, the most well-studied lantibiotic nisin interacts with the pyrophosphate fragment of lipid II. Critical to this binding are the two N-terminal rings of the lantibiotic [88]. The formation of the pore complex results in cell membrane permeabilization and dissipation of the proton motive force [87].

In general, lantibiotics are synthesized and secreted by Gram-positive microorganisms, and their activity is most often manifested in connection with closely related Gram-positive bacteria. For Gram-negative bacteria, their activity is rather limited [86] because the cell wall of Gram-negative bacteria is an effective permeability barrier due to the presence of an outer membrane, which prevents access to the peptidoglycan layer (localization of lipid II) and the cytoplasmic membrane (Figure 2). Moreover, the anionic cell surface of Gram-negative bacteria promotes the binding of cationic lantibiotics, where such an interaction potentially increases the stability of the outer membrane through electrostatic interactions [89].

*B. licheniformis* produces two lantibiotics that may be important for applications in various industries (Table 1).

The first one is sublichenin, which is a subtilin-like lantibiotic of probiotic bacterium *Bacillus licheniformis* MCC 2512 that has a molecular weight of 3.348 kDa and the succinylated form has a molecular weight of 3.448 kDa [31,49]. Subtilin is a natural variant of nisin that refers to linear pentacyclic class I antibiotics [90]. The sublichenins from *B. licheniformis* are almost identical to the JS-4 subtilin from *B. subtilis*. Subtilin JS-4 retained >90% and 86.1% of its antibacterial activity even after a 30 min exposure to 80–100 °C and 121 °C, respectively, indicating considerable thermostability. Subtilin JS-4 was also rapidly inactivated by proteolytic enzymes including proteinase K, trypsin, papain and pepsin [91]. It also showed a broad antimicrobial spectrum against Gram-positive bacteria. Subtilin JS-4 inhibited the growth of foodborne bacterium *L. monocytogenes* by increasing cell membrane permeability, triggering pore formation and K^+^ leakage, and damaging cell membrane integrity, which eventually disrupted the membrane and cellular structure [91] (Figure 2).

The second antibiotic is lichenicidin, a dipeptide lantibiotic consisting of a synergetic lantibiotic pair, Licα (3.251kDa) and Licβ (3.021 kDa), which was described for *B. licheniformis* DSM 13. This substance demonstrated activity against the growth of Gram-positive bacteria, such as *Bacillus subtilis*, *Micrococcus luteus*, *Staphylococcus aureus*, *Streptococcus pyogenes*, *Staphylococcus simulans* and *enterococci* but neither caused hemolysis nor inhibited the growth of Gram-negative bacteria. Lichenicidin is associated with the cell surface and shows stability against trypsin, chymotrypsin, and proteases [50]. Moreover, lichenicidin can be produced by other strains of *B. licheniformis*, and the structure of its peptides may differ depending on the strain. Lichenicidin was not cytotoxic to human erythrocytes and fibroblasts [92]. *B licheniformis* strain ATCC 14580 produced lichenicidin with activity against a range of pathogenic microorganisms including *Listeria monocytogenes*, *Staphylococcus aureus*, vancomycin-resistant enterococci, *Bacillus cereus*, *Streptococcus pneumoniae* and *Streptococcus* mutants [51]. Lichenicidin has been produced by *B. licheniformis* strain VK21 [52], and WIT 562, 564 and 566 [53]. Additionally, lichenicidin production was found in *B. licheniformis* isolates (isolated from retail infant milk formulae): strains IMF20, IMF66, IMF69 and IMF80. These strains demonstrated antimicrobial activity against target Gram-positive organisms. No activity was observed against the Gram-negative bacteria *E. coli* or *S. typhimurium* [54]. For Lchα and Lchβ, lichenicidin subunits produced by VK21 strain, described tertiary structures and details of mode of action [52].

Another variant of lichenicidin comprises two mature peptides, Bliα and Bliβ, produced by the I89 strain; their synergistic activity is required for full activity [55,93].

Lichenicidin acts through a dual mode of action that involves the α subunit recognition of lipid II, providing specificity and stability for the interaction of β subunit, which induces leakage of the intracellular contents of bacteria [94,95] (Figure 2).

#### 2.1.2. Class II: Heat-Stable Non-Lanthionine Peptides Smaller Than 10 kDa

This class includes unmodified peptides with a molecular weight up to 10 kDa. The vast majority of them are thermostable membrane-active peptides. Among them, peptides that are active only against Gram-positive microorganisms and active against both Gram-positive and Gram-negative microorganisms can be distinguished (Table 1). Peptides with antifungal (Table 2) and amoebolytic (Table 3) activities have also been identified.

##### *B. licheniformis* Secreted Peptides Active Only against Gram-Positive Microorganisms

Because antagonism provides a survival advantage in the suppression of related species of microorganisms, most bacteriocins secreted by different strains of *B. licheniformis* are active only against Gram-positive bacteria (Table 1). Peptides may be insensitive or sensitive to the action of proteolytic enzymes. However, the vast majority of the identified bacteriocins that are active only against Gram-positive microorganisms are sensitive to the action of proteinases.

Bacillocin 490, a bacteriocin with a low molecular mass (2 kDa) and produced by a thermophilic strain (*B. licheniformis* 490/5) isolated from dairy foods, shows high thermal stability, with 46.4% residual activity after 1 h of exposure to 100 °C. This bacteriocin was inactivated by pronase E and proteinase K. Bactericidal activity was maintained during storage at 4 °C and was remarkably stable over a wide pH range. The activity range of bacillocin 490 was limited to some Gram-positive bacteria. The highest antimicrobial activity was observed against *Bacillus stearothermophilus*, *B. smithii*, *B. subtilis* and *B. anthracis*. Moderate inhibition of *B. cereus*, very low inhibition of *Listeria innocua* and *S. aureus*, and no inhibition of *B. thuringensis* and *Streptococcus thermophilus* were noted. This activity spectrum shows that bacillocin 490 is principally active against species phylogenetically related to the producer strain. The incubation of *B. smithii* in the presence of bacillocin 490 resulted in 96% death in 30 min, indicating that bacteriocin has a bactericidal effect [28].

The supernatant of thermophilic strain *B. licheniformis* H1 exhibited antagonistic activity against various species of Gram-positive bacteria, such as *Listeria monocytogenes* but not against Gram-negative bacteria, except for *Pseudomonas fluorescens*. The inactivation of this bacteriocin-like activity by a-chymotrypsin, trypsin, and papain was highly significant. No significant decrease was found in antimicrobial activity after the incubation of a bacteriocin-containing supernatant from *B. licheniformis* H1with pepsin or lipase. The bacteriocin-like substance was stable at temperatures up to 75 °C for 60 min, but activity was lost after being autoclaved at 121 °C for 15 min. Concentrated antimicrobial activity was found in the protein fraction obtained with 60% ammonium sulfate saturation. The results of sodium dodecyl sulfate–polyacrylamide electrophoresis analysis of concentrated partially purified supernatants collected after resting bacterial cells at 55 °C revealed a bacteriocin-like protein with a molecular mass of approximately 3.5 kDa [56].

*B. licheniformis* AnBa9 produced antibacterial bacteriocin-type peptides with a molecular mass of <10 kDa. The production of these peptides was 25-fold higher under optimized conditions for producer growth compared with nonoptimized condition. The level of bacteriocin production and its specific activity gradually decreased with increasing concentrations of lactose and NH_4_NO_3_. F high concentration of yeast extract, an alkaline pH and an elevated temperature improved the production of antibacterial peptides by *B. licheniformis* AnBa9. *B. licheniformis* AnBa9 inhibited several Gram-positive bacteria, including *Staphylococcus aureus*, *Bacillus cereus*, *Staphylococcus epidermidis*, *Kurthia gibsonii*, *Micrococcus luteus*, *Streptococcus mitis*, *Bacillus subtilis*, *L. lactis*, *Bacillus smithii*, *Lactobacillus acidophilus*, *Pediococcus acidilactici*, and *Leuconostoc mesenteriodes*. However, these bacteriocins did not inhibit *Listeria* strains or Gram-negative bacteria. The loss of antibacterial activity of the permeate after treatment with proteinase K, pronase E, and trypsin, suggested that these bacteriocins are sensitive to proteolytic enzymes. They were resistant to temperature up to 100 °C for 30 min and over a wide range of pH from 4 to 12 [35].

Under anaerobic conditions, *B. licheniformis* 26L10/3RA produced an inhibitory bacteriocin-like component called lichenin. This peptide was purified to homogeneity, and the molecular mass was estimated at approximately 1.4 kDa. Lichenin was found to be hydrophobic, was sensitive to atmospheric oxygen, retained biological activity even after boiling for 10 min, and was active over a pH range of 4.0–9.0. It was active against *Streptococcus bovis*, *Ruminococcus albus*, *Ruminococcus avefaciens*, and *Eubacterium ruminantium*. The biological activity of this peptide was completely inactivated by proteinase K treatment, but this peptide was resistant to trypsin. Heat treatment at 80 °C and boiling for 10 min at pH 4.0 and 9.0 resulted in a significant reduction in biological activity. Lichenin production was observed only upon *B. licheniformis* anaerobic growth, the antibacterial activity of which was also demonstrated for the reference strains grown under anaerobic conditions only. The inability of lichenin to inhibit aerobically grown bacteria was explained either by its inactivation by atmospheric oxygen or the target bacteria due to oxidative respiration. No N-terminal block was observed in the sequence and the peptide did not show any characteristics of cyclicity. However, the seventh amino acid residue could not be identified and it did not belong to any of the natural amino acids [32].

Strain BTHT8, identified as *B. licheniformis*, inhibited the growth of Gram-positive test organisms. The active component, labeled as bacteriocin BL8, was purified from the supernatant of *B. licheniformis* strain BTHT8. The molecular mass was determined as 1.4 kDa. The results of N-terminal amino acid sequencing of BL8 identified a 13 amino acid sequence stretch. Bacteriocin BL8 was stable even after boiling at 100 °C for 30 min and over a wide pH range of 1–12 [57].

A bacteriocin from *B. licheniformis* cy2, named BSCY2, was stable at pH 2.5–9.5, showing activity against *B. subtilis*. BSCY2 was stable below 40 °C and retained its antimicrobial activity during long tern storage at −20 °C and −70 °C. BSCY2 was inactivated after 15 min exposure to temperatures over 80 °C and lost 50% of its antimicrobial activity within 2 h at 70 °C. BSCY2 was inactivated by proteinase K treatment, which indicated its proteinous nature. The direct detection of the BSCY2 band showing antimicrobial activity on Tricine–SDS-PAGE suggested an apparent molecular mass of about 6.5 kDa [58].

In contrast to the above-mentioned bacteriocins of this group, some bacteriocins retain their activity after treatment with proteolytic enzymes.

*B. licheniformis* strain VPS50.2 produced bacteriocin licheniocin 50.2 (molecular mass of approximately 3.25 kDa) and was effective against Gram-positive bacteria, including *Listeria monocytogenes*, methicillin-resistant *Staphylococcus aureus*, and b-haemolytic streptococci. Bacteriocin activity was insensitive to lysozyme and proteinase K, being heat stable after incubation at 100 °C for 30 min and over a wide range of pH (2–12). The inhibitory spectrum considered in this study was limited to Gram-positive bacteria only. The maximum antagonistic activity was found in the precipitate with 60% saturation of ammonium sulfate [59].

Despite the varying degrees of sensitivity to the action of proteolytic enzymes, bacteriocins of this group are resistant to elevated temperatures and wide pH values, so they are especially suitable for medical applications.

##### *B. licheniformis* Secreted Peptides Active against Both Gram-Positive and Gram-Negative Microorganisms

The bacteriocins secreted by *B. licheniformis* that show activity against both Gram-positive and Gram-negative microorganisms are common and are sensitive to the action of proteolytic enzymes but resistant to elevated temperatures. They show different sensitivities to a wide range of pH values. All bacteriocins of this sub-group (Table 1) are sensitive to the action of proteinases.

*B. licheniformis* strain IITRHR2 produced a bacteriocin-like inhibitory substance (~1.2 kDa), which was thermostable (up to 80 °C but showed decreased activity at higher temperatures) and pH-resistant but lost activity when subjected to proteinase treatment (proteinase K and pronase E). This bacteriocin inhibited various Gram-positive bacterial strains, such as *B. subtilis*, *B. cereus*, *Streptococcus thermophilus*, *Pediococcus pentosaceus*, *Leuconostoc mesenteroides*, *L. monocytogenes*, *Bifidobacterium bifidum*, and *Enterococcus faecalis*. The growth of Gram-negative bacteria *Shigella flexneri*, *Shigella sonnei* and *Pseudomonas aeruginosa* was also inhibited by this compound [60].

A culture supernatant of *B. licheniformis* MKU3 exhibited bacteriocin-like activity against several type strains of Gram-positive bacteria, such as *Bacillus subtilis*, *Bacillus smithii*, *Staphylococcus epidermidis*, *Micrococcus luteus*, *Leuconostoc mesenteriodes* and *Pediococus acidilactici*, *B. cereus*, *B. megaterium*, *K. gibsonii*, *Staphyloccus* sp., *Streptococcus* sp., and *Micrococcus caseolyticus* but not *Listeria* sp. However, Gram-negative bacteria, such as *Serratia marcescens* and *Pseudomonas fluorescens* B10 were not inhibited by this bacteriocin, excluding *Escherichia coli*. The extract showed strong activity against different fungi including *Aspergillus niger*, *A. versicolor*, *A. fischeri*, and *A. fumigatus* and the yeast *Candida albicans*. The active substance apparently is a bacteriocin-like protein with a molecular mass of 1.5 kDa. The activity of this bacteriocin was stable at pH 3.0–10.0 and temperatures up to 100 °C for 60 min but was inactivated by proteinase K, trypsin or pronase E. The bacteriocin lost its activity after incubation at 121 °C for 15 min. The composition of the medium affected the production of this bacteriocin [34].

*B. licheniformis* strain B116 showed strong antimicrobial activity against *Staphylococcus aureus* and *Salmonella enterica ser. Pullorum*. Bacteriocin was precipitated by ammonium sulfate, and its molecular mass was determined as ~4 kDa. The culture supernatant of this strain exhibited antimicrobial activity against both Gram-positive and Gram-negative bacteria, including *Bacillus cereus*, *Staphylococcus aureus*, *Listeria monocytogenes*, *Micrococcus luteus*, *Escherichia coli*, *Streptococcus equi*, and *Salmonella* spp. The bacteriocin was resistant to heat, acid, and alkaline treatments. The activity of this bacteriocin was totally lost after digestion by pronase and activity was partially lost after digestion by papain and lipase. The inactivation by lipase indicated that this bacteriocin may have contained a lipid moiety [61].

*B. licheniformis* MCC 2016 (strain was also named Me1) produced the antibacterial peptide ppABP that was completely abolished by proteinase K. The culture, which was isolated from milk, was able to produce a proteinaceous antibacterial peptide with a low molecular weight between 3.0 and 3.5 kDa. It exhibited a broad spectrum of inhibitory activity and was stable over a wide range of temperature and pH. ppABPs were found to be thermally stable for 15 min at 80 °C. The SN of this strain exhibited inhibitory activity against both Gram-positive and Gram-negative food-borne and human pathogens [63,64]. Films activated with ppABP from *B. licheniformis* Me1 showed a zone of inhibition that was not confined to the film area, indicating that the ppABP diffused from the films into the medium [62].

*B. licheniformis* strain N.C.T.C. 7072 produced licheniformins, which are antibacterial agents, with in vitro bacteriostatic activity against many organisms, including *Mycobacterium tuberculosis*. In addition to inhibiting the growth of mycobacteria, they showed efficacy against *Staphylococcus aureus* and *Escherichia coli* [65]. The peptides had a molecular mass of 3.8, 4.4 and 4.8 kDa, respectively [37].

*B. licheniformis* strains MCC2512 and MCC2514 exhibited inhibitory activity against *Micrococcus luteus*, *Staphylococcus aureus*, *Klebsiella* sp., and *Aeromonas hydrophila*. In addition to these pathogenic strains, B. *licheniformis* strain MCC2512 also showed inhibitory activity against *Listeria monocytogenes* and *Salmonella typhimurium*. The activities of the bacteriocins from both cultures were completely lost upon exposure to proteinase K, indicating the proteinaceous nature of the compounds. After treating the sample with trypsin and pepsin, 100% of the activity was retained; however, with a-amylase, 50% of the activity was lost. The isolated bacteriocins varied in their mechanisms of action and stability. The molecular weights of the inhibitor components from MCC2514 and MCC2512 were 6.5 and 3.5 kDa, respectively. *B. licheniformis* MCC2512 produced a subtilin-type antimicrobial compound that acted on cell wall synthesis. Strain MCC2514 inhibited RNA synthesis [31]. The active substance produced by *B. licheniformis* MCC2512 was identified as sublichenin [49] (Figure 2).

##### *B. licheniformis* Peptide Activity against Fungal Pathogens

An important characteristic of some bacteriocins is antifungal activity (Table 2), which substantially expands the horizons of their application in medicine, agriculture, and the food industry.

The cell-free supernatant of *B. licheniformis,* ZJU12, isolated from soil exhibited pronounced antibacterial (for Gram-positive bacteria) activity. The bacteriocin-like peptides produced by *B. licheniformis* ZJU12 showed no activity against Gram-negative bacteria, but exhibited activity against fungi (*Xanthomonas oryzae pv.oryzae*, *Alternaria brassicae*, *Fusarium oxysporum*, and others). After treatment with proteinase K and trypsin, the antagonistic activity was completely lost. The molecular mass estimated via Tricine–SDS-PAGE of the antagonistic compound was approximately 3 kDa. These characteristics indicated that the antagonistic substances produced by this strain had the property of bacteriocin. The activity was stable following exposure up to 100 °C for 30 min but was completely lost at 121 °C for 15 min. The maximum antagonistic activity was found in the resolved precipitate of supernatant with 60% saturation of ammonium sulfate. The toxicity of the substance was low because no adverse effects on mice were detected at a dose of up to 0.8 mg/20 g in acute toxicity tests [33].

*B. licheniformis* strain MGrP1 produced antibiotics in liquid media containing soyabean meal and mannitol that inhibited the growth of plant fungal pathogens of agricultural importance, namely *Colletotrichum lindemuthianum* (bean anthracnose), *Colletotrichum kahawae* (coffee berry disease), *Fusarium oxysporum *f.sp.* phaseoli* (fusarium yellow), and *Alternaria solani* (early blight). The results of paper chromatography combined with bioautography revealed two thermostable active compounds whose activity was optimal at pH 6. Low pH ranges and autoclaving temperatures significantly reduced the activities of the antibiotics [96].

Fungicin M-4, produced by *B. licheniformis* M-4, is composed of 34 amino acid residues of 7 different amino acids, including 4 residues of ornithine per molecule. The same strain showed inhibitory activity against the human pathogenic amoeba *Naegleria fowleri*. Purified fungicin M-4 demonstrated antifungal activity against the pathogenic fungi *Sporothrix schenckii* and *Microsporum canis.* Fungicin M-4 was resistant to proteolytic enzymes and lipase. Its antifungal activity was fairly resistant to heat, although incubation at 80 °C for 30 min caused 30% inactivation. The activity was stable in the pH range from 2.5 to 9.0. Its molecular weight was 3.6 kDa. Attempts to deduce an amino acid sequence were unsuccessful, suggesting that fungicin may be a cyclic peptide or it is blocked at its amino-terminal end [97].

Peptide A12-C from *B. licheniformis* A12 showed a pronounced antifungal effect, being an acidic hydrophilic peptide with a mass of 0.77 kDa, containing only six different amino acids. Peptide A12-C was resistant to proteolytic enzymes, such as trypsin, pronase, and proteinase K, as well as to carboxypeptidase A, alkaline phosphatase, lipase, lysozyme, β-glucosidase, and β-glucuronidase. Peptide A12-C showed resistance to heat (100 °C for 30 min at pH 7.0) and incubation at room temperature under acidic conditions (pH 2.5), but 75% of the activity was lost after incubation at pH 9.5 for 30 min at room temperature. Peptide A12-C was active against several fungi (*Microsporum canis*, *Mucor mucedo*, *M. plumbeus*, *Sporothrix schenckii*, and *Trichophyton mentagrophytes*) and bacteria (*Bacillus megaterium*, *Corynebacteriurn glutamicum*, *Sarcina* and *Mycobacterium phlei*) [46].

*B. licheniformis* NCIMB 8874 produced peptide ComX with antifungal activity against the fungal leaf pathogen *Alternaria alternata*. ComX comprises 13 amino acid residues: Glu-Ala-Gly-Trp-Gly-Pro-Tyr-Pro-Asn-Leu-Trp-Phe-Lys [99].

##### Amoebolytic Substances from *B. licheniformis*

Bacteriocins with amoebolytic activity (Table 3) have been identified, all of which showed resistance to the action of proteolytic enzymes and elevated temperatures.

*B. licheniformis* A12 produced two amoebolytic substances (amoebicins A12-A and A12-B) in liquid media during sporulation. Both substances were heat- and protease-resistant peptides containing aspartic acid, glutamic acid, serine, proline, and tyrosine in a molar ratio of 5:2:2:2:2. No fatty acids or carbohydrates were detected. Both amoebicins retained 100% of their activity after being heated at 100 °C for 30 min at pH 7.0. They were also resistant to incubation at room temperature under acidic conditions (pH 2.5) but lost 75% of their activity upon incubation at pH 9.5 for 30 min. The crude supernatants, as well as the purified substances, retained 100% of their activity after storage for 1 month at 4 °C or for 6 months at −20 °C. Amoebicins A12-A and A12-B were resistant to the enzymes trypsin, pronase, proteinase K, alkaline phosphatase, lipase, lysozyme, α-glucosidase, and 3-glucuronidase. They were also resistant to carboxy peptidase A, suggesting that a free carboxyl terminus was not present. Their molecular weight was found to be 1.43–1.60 kDa. Purified amoebicins A12-A and A12-B exhibited amoebolytic action against *Naegleria fowleri*. They also exhibited antibiotic action against yeasts (*Saccharomyces heterogenicus* and *Cryptococcus neoformans*) and several fungal species (*Aspergillus niger*, *Microsporum canis*, *Mucor plumbeus*, and *Trychophyton mentagrophytes*). Their antibacterial spectrum appears to be restricted to *Bacillus megaterium*, *Corynebacterium glutamicum*, and *Sarcina* sp. The amoebolytic effect was studied via electron microscopy. At 10 min after addition, the characteristic shape of the cells changed. First, they developed abnormal globular pseudopodia, and then they became rounded. After 30 min of incubation, the cell membrane ruptured, with the release of the cytoplasmic material. All of this was followed by complete cellular destruction within 1 h [107].

*B. licheniformis* M-4 produced three antibiotic peptides (m4-A, m4-B, and m4-C) with amoebolytic activity. They were active against human pathogenic and non-pathogenic strains of *Naegleria fowleri*, which is the causative agent of primary amoebic meningoencephalitis. The amoebicidal activity of these peptides was resistant to the actions of trypsin, proteinase K, and carboxypeptidase A. They were cyclic peptides with molecular weights ranging from 3.0 to 3.2 kDa. These peptides were composed of six different amino acids (Asp, Glu, Ser, Thr, Pro, and Tyr), and they differed only in the number of Asp residues. The three amoebicins had a broad antifungal spectrum, although peptide m4-C showed a two-fold higher specific activity against a variety of fungi and yeasts than others. The three peptides showed a narrow antibacterial spectrum, but *Bacillus megaterium* (not spores) was highly sensitive [98]. The amoebicins from *B. licheniformis* M-4 differ from those produced by strain A12 in molecular weight, in their amino acid composition (A12-A and A12-B contained threonine), in the number of residues per molecule, and in their solubility in water (A12-A and A12-B are not water-soluble) [98,107].

*B. licheniformis* D-13 produced three hydrophobic peptides (amoebicins d13-A, d13-B, and d13-C) that showed anti-amoebic activity against human-pathogenic and non-pathogenic species of Naegleria and had a broad spectrum of antibacterial activity. The three amoebicins showed the same amino acid composition and molecular weight of 1.87 kDa. The three amoebicins were stable at pH 2.5 to 9.5, and they retained 100% of their activity after being heated at 100 °C for 30 min and after being stored at −20 °C for 6 months. Because the purified amoebicins were not soluble in aqueous buffers, a mixture of partially purified amoebicins in 20 mM Tris-HCl (pH 7.2) was tested for sensitivity to various enzymes. The mixture retained 100% of its activity after being treated for 1 h with proteases (trypsin, pronase, and proteinase K), lipase, or β-glucuronidase. Amoebicin d13-B caused the lysis of amoebae through the disorganization of the cell membrane (Figure 2). No amino acid residues were detected after the N-terminal sequence of amoebicin d13-B, suggesting that this peptide is cyclic or blocked at its amino terminus [108].

#### 2.1.3. Class III: Heat-Stable and Heat-Labile Proteins Larger Than 10 kDa

This class includes unmodified peptides with a molecular weight larger than 10 kDa (Table 1). In most cases, these are thermostable membrane-active peptides that are sensitive to proteinase treatment.

*B. licheniformis* SXAU06 produced a bacteriocin-like substance (BLIS) with an approximate molecular weight of 14 kDa designated as BLIS_SXAU06. It was active against *Escherichia coli*, *Salmonella enterica*, *Staphylococcus aureus*, *Staphylococcus epidermidis*, *Micrococcus luteus*, and *Listeria monocytogenes.* BLIS_SXAU06 exhibited high resistance to treatment at high temperature, high acidity and alkalinity, and proteinase K, but it was fully inactivated by pronase E and partially inactivated by trypsin and pepsin. BLIS_SXAU06 was heterologously expressed in *E. coli*, and the recombinant BLIS_SXAU06 exhibited effective antibacterial activity against *S. aureus*, *S. epidermidis*, *M. luteus*, and *L. monocytogenes* [66].

When a tropical marine strain of *B. licheniformis* D1 was grown in Luria–Bertani (LB) broth-containing tryptone medium, it produced a 14 kDa protein BL-DZ1 (BL00275) with antimicrobial activity against pathogenic *Candida albicans* BH, *Pseudomonas aeruginosa* PAO1, and biofouling *Bacillus pumilus* TiO1 cultures. The antimicrobial activity was lost after treatment with trypsin and proteinase K. The protein was stable at 75 °C for 30 min and over a pH range of 3.0 to 11.0. The BL-DZ1 protein was able to inhibit both biofilm growth and disrupt pre-formed biofilms of *C. albicans*, *P. aeruginosa*, and *B. pumilus* [67].

*B. licheniformis* HS10 produced an antifungal protein with a molecular weight of approximately 55 kDa, identified as carboxypeptidase. It showed significant inhibition activity of eight different kinds of plant pathogenic fungi, and it was stable with strong biological activity at as high as 100 °C for 30 min and in pH values ranging from 6 to 10. The biological activity was negatively affected by protease K. The protein had broad spectrum antifungal activity against seven kinds of plant pathogenic fungi [100].

Isolated from seaweed, *B. licheniformis* produced a protein with antibacterial activity against methicillin-resistant *Staphylococcus aureus*, vancomycin-resistant enterococci, and *Listeria monocytogenes.* The antibacterial activity was strongest in cultures grown under shaking at 210 to 230 rpm. No antibacterial activity was found in cultures grown statically or at other rotary shaking speeds. The antibacterial compound was sensitive to proteinase K, pronase, and trypsin, but was not affected by Tween-20, -40, -60, or -80, or a- or b-amylase. Its activity was not adversely affected by heating up to 40 °C or treatment at pH from 5 to 14. The bioactive compound was determined to be associated with a 30.7 kDa protein, that showed homology to the secreted YbdN protein of *B. licheniformis* ATCC 14580 [68].

*B. licheniformis* MY75 secreted high levels of extracellular chitinase with a molecular weight of 55 kDa and inhibited the growth of pathogenic fungi *Gibberella saubinetii* and *Aspergillus niger.* The secretion of this protein was induced by chitin powder [69]. Chitinase proteins were present in the culture supernatant of *B. licheniformis* Mb-2 [74], *B. licheniformis* TP–1 [75], *B. licheniformis* S213 [70], *B. licheniformis* SSCL-10 [71], *B. licheniformis* B307 [72].

*B. licheniformis* BS-3 produced an antifungal 31 kDa protein, F2, that inhibited the growth of *Aspergillus niger*, *Magnaporthe oryzae*, and *Rhizoctonia solani.* The F2 protein was moderately resistant to hydrolysis by trypsin and proteinase K. Higher F2 antifungal activity was observed from pH 6.0 to pH 10.0 and at a temperature below 70 °C for 30 min [76].

As in the other cases, this group of bacteriocins contains some proteinase-resistant members. Owing to this property, these proteins may be applicable for administration through the digestive system.

*B. licheniformis* strain JS produced 16 kDa antimicrobial protein (AMP), which demonstrated more activity against Gram-positive bacteria *Bacillus cereus* and less activity against Gram-negative bacteria (*S. dysenteriae* and *S. typhimurium*). The purified peptide also increased the effectiveness of antibiotics, such as kanamycin, neomycin, and streptomycin. Hence, it could be important because the AMPs produced by *B. licheniformis* may facilitate the entry of these antibiotics inside the pathogens and increase their efficiency. The antimicrobial activity was 100% after AMP incubation between 10 and 90 °C. The trypsin digestion study revealed that AMP retained 100% of its activity [77].

*B. licheniformis* T6-5 inhibited more than 65% of the 40 *Bacillus* strains and of the sulfate-reducing bacteria *Desulfovibrio alaskensis*. The treatment of the supernatant with organic solvents led to total (acetone, ethanol, and methanol) or partial (chloroform) inactivation of the inhibitor component. The inhibitor probably contained a lipidic portion as a part of its structure. This substance was heat-stable after incubation at 100 °C for 1 h and maintained its activity after being autoclaved at 121 °C for 15 min. It was active in a wide range of pH (3.5–9.5). The inhibitory component was resistant to the action of pronase E, proteinase K, trypsin, RNase, chitinase, b-galactosidase, a-galactosidase, and manosidase. The substance produced by strain T6-5 was estimated via dialysis to be bigger than 12 kDa. According to the results of SDS-PAGE analysis, strain T6-5 showed an inhibitory zone of ca. 20 kDa, corresponding to the molecular weight suggested by the dialysis membrane approach [36]. The substance inhibitory zones of *B. licheniformis* H2O-1 antimicrobial were related to a region of high molecular mass (90–120 kDa) [36]. *B. licheniformis* strains T6-5 and H2O-1 prevented the formation of *B. pumilus* LF4 biofilm and eliminated pre-established LF4 biofilm [78]. The nature and precise structure of the above inhibitory substances are still unclear.

#### 2.1.4. Class IV: Complex with a Lipid Moiety or Carbohydrate Moiety

*B. licheniformis* BFP011 isolated from papaya (Thailand), produced extracellular antimicrobial substances that were active against some important phytopathogens, pathogenic and spoilage microorganisms, such as *Colletotrichum capsici*, and *Escherichia coli* O157: H7, and *Salmonella typhi* ATCC 5784. The three types of antimicrobial substances (F4, F5, and F6) produced by *B. licheniformis* BFP011 were not sensitive to pronase as revealed in stationary phase cultures. The antimicrobial substances of this bacterium were stable at 37 and 70 °C and partly resistant at 121 °C. Most of the antimicrobial protein substances from the culture supernatant were extracellular compounds having low molecular weights of less than 45 kDa. The antimicrobial substances of *B. licheniformis* BFP011 contained peptides and unsaturated fatty acids; however, the precise structural organizations of these compounds are not known. They exhibited a broad spectrum of antimicrobial activity against both Gram-positive and Gram-negative bacteria and the fungus *C. capsici*. These substances differed from iturin A (commercial), bacitracin (commercial), and a bacteriocin-like substance of *B. licheniformis* P40 [79] (Table 1).

Two glycolipopeptides, ieodoglucomides A and B, were isolated from marine-derived *Bacillus licheniformis* 09IDYM23. They consisted of an amino acid, a new fatty acid, a succinic acid, and a sugar. These glycolipopeptides showed moderate antimicrobial activity when tested against both Gram-positive and Gram-negative bacteria and fungi, such as *S. aureus*, *P. aeruginosa*, *E. coli*, *B. cereus*, and *A. niger.* The molecular formula of ieodoglucomides A and B were assigned as C_30_H_53_NO_12_ and –C_29_H_51_NO_12,_ respectively [80].

The same strain, 09IDYM23, produced a glycolipopeptide, ieodoglucomide C, and a new monoacyldiglycosylglycerolipid, ieodoglycolipid. These compounds showed antimicrobial activity against fungi *C. albicans*, *A. niger*, *R. solani*, *C. acutatum*, and *B. cenerea* and bacteria *S. aureus*, *B. subtilis*, *B. cereus*, *S. typhi*, *E. coli*, and *P. aeruginosa*. The molecular formulae of each isolated component were determined to be C_29_H_51_NO_12_ and C_30_H_56_O_14,_ respectively [81].

#### 2.1.5. Class V: Bacteriocins with Undetermined Molecular Weight

A skin isolate of *B. licheniformis* showed the most potent antibacterial activity at pH 7, with an incubation period of 48 h, at an incubation temperature of 25 °C. The antipathogenic metabolites were then detected as bacteriocin-like substances, which demonstrated heat stability up to 80 °C for 30 min. The papain-treated cell-free supernatant did not show any bacteriocin activity, suggesting that the substances were antimicrobial peptides. This bacteriocin inhibited the growth of *Staph. aureus* and *Kl. pneumoniae *subsp.* Pneumonia* [109] (Table 1).

A skin isolate *B. licheniformis* UpA was observed to produce antimicrobial metabolite that was effective against *Klebsiella pneumoniae *subsp.* pneumoniae*. It was detected as a bacteriocin-like substance and was further confirmed as an antimicrobial peptide through papain treatment. The produced bacteriocin was stable with heat treatment up to 80 °C for 30 min and up to pH 7 [83].

The supernatant of *B. licheniformis* A-1-5B-AP significantly reduced the growth of oral pathogenic strains *Porphyromonas gulae* 3/H, *Prevotella intermedia* 1/P, and *Streptococcus mutans* ATCC 35668. However, *B. licheniformis* A-2-11B-AP only significantly inhibited the growth of *P. intermedia* 1/P and *S. mutans* ATCC 35668. The enzyme-treated SN of *B. licheniformis* A-1-5B-AP did not lose its antimicrobial effect and significantly inhibited the growth of *Micrococcus luteus* DSM 1790. Proteinase K, lipase, or α-amylase did not affect the antimicrobial activity present in the SN of strain of *B. licheniformis* A-1-5B-AP. The presence of genes associated with the synthesis of lichenysin was detected, although their presence in the medium was not confirmed [84].

*B. licheniformis* HJ2020 MT192715.1 produced a bacteriocin active against many food spoilage microorganisms. The residual inhibition activity of bacteriocin varied according to the incubation conditions and treatment type. The inhibitory activity was 220 and 360 U mL^−1^ against the clinical isolates of pathogenic strains *Escherichia coli* and *Salmonella typhi*, respectively; the activity was 42, 60, and 80 U/mL against to *B. subtilis*, *B. cereus* and *Candida albican,* respectively [85]. No activity was detected against *Lactobacillus* or *Bifidobacterium*. These results are similar to those reported for *B. licheniformis* P40 [29]. The bacteriocin lost approximately 25–40% of its activity when incubated in acidic pH (between 3 and 5), it lost approximately 80% of its activity at pH 10, and no activity was observed at pH 12. The heat stability of the bacteriocin also was tested, and the results showed that it retained all activity when incubated at 5–35 °C for 30 min. It lost approximately 25–50% of its activity after incubation at 50–80 °C and lost all activity when incubated at 100 °C for 30 min or autoclaved at 121 °C for 15 min at 15 psi. The reduction in bacteriocin activity and the loss of all of its activity at high temperature were attributed to denaturation, indicating the proteinaceous nature of bacteriocin. The results also revealed that the bacteriocin was stable when treated with α-amylase and lipase, pointing to the absence of glycosidic or lipidic residuals [85].

The bacteriocins produced by *B. licheniformis* are characterized by resistance to various pH ranges, thermal stability, and, in some cases, sensitivity to proteolytic enzymes. However, they differ in the spectrum of antibacterial activity for different strains of *B. licheniformis*. For example, a bacteriocin produced by *B. licheniformis* MKU3 isolated from slaughterhouse sediments did not inhibit *L. monocytogenes*, *P. fluorescens* or *S. marcescens*, but inhibited *E. coli* [34]. A bacteriocin-like peptide produced by *B. licheniformis* ZJU12 isolated from soil exhibited antagonistic activity against *S. aureus* [33]; *B. licheniformis* P40 inhibited *E. aerogenes* but did not inhibit *P. fluorescens* [29]. Anaerobiosis specifically expressing lichenin demonstrated a narrow spectrum of activity against ruminal anaerobes [32].

Bacteriocins have different structures and different mechanisms of action against bacteria. Their ability to reach their targets is crucial for their effectiveness. Due to variations in cell wall structure, certain bacteriocins, particularly those designed to target intracellular components, may encounter challenges in penetrating the cell walls of mycobacteria or Gram-negative bacteria. Conversely, pore-forming bacteriocins have demonstrated a broader spectrum of activity against various types of bacteria.

### 2.2. Non-Ribosomal Biosynthesized Peptides

Non-ribosomal peptides (Table 4) are synthesized via the sequential condensation of amino acids by special non-ribosomal multimodular peptide synthetases, which are mainly found in bacteria and fungi. Many peptides, not produced by ribosomes, contain unnatural amino acids and other molecules that are not found in the peptides synthesized by ribosomes [110]. Such peptides include many well-known substances, such as antibacterial drugs (penicillin and vancomycin), antitumor compounds (bleomycin) and immunosuppressants (cyclosporine) [111].

#### 2.2.1. Bacitracin

Bacitracin, the first non-ribosomal peptide antibiotic isolated from *B. licheniformis* cultures [130], is actively used in human and veterinary medicine and shows sufficient safety [131]. It is a topical medicine used for disinfection of wound. Bacitracin is a polypeptide of approximately 1.42 kDa, and a non-ribosomally synthesized dodecapeptide antibiotic produced by certain strains of *B. subtilis* and *B. licheniformis* [114]. Bacitracin contains 12 amino acids, 4 of which are the D-isomers of glutamic acid, aspartic acid, phenylalanine, and ornithine [132]. The synthesis of this peptide is rare in other species of the genus *Bacillus*, which indicates the importance of its discovery in *B. licheniformis*. This antibiotic inhibits the cell wall synthesis of many Gram-positive and some Gram-negative bacteria [133]. In addition, due to its fast elimination rate and low absorption, it can be used as an additive in animal feed [134]. The bacitracin from *B. licheniformis* is also known as ayficin [113] (Table 4). This antibiotic is a mixture of at least five polypeptides and consists of three separate compounds: bacitracin A, B, and C [135]. This antibiotic is released from bacteria only under cultural conditions that eventually support spore formation [132]. Bacitracin begins to be synthesized in the early exponential phase of vegetative growth, reaching a constant rate in the stationary phase of growth in synthetic media without glucose. The addition of glucose inhibited the synthesis of bacitracin; however, this inhibition was not the result of catabolite repression but occurred owing to a decrease in the pH of the growth medium, presumably due to the accumulation of pyruvate and acetate [115].

Bacitracin showed a potent antibiotic activity against Gram-positive cocci, staphylococci, streptococci, corynebacteria, *Treponema pallidum*, *T. vincenti*, *Actinomyces israeli*, anaerobic cocci, clostridia, neisseria, most gonococci, and meningococci, but it was relatively ineffective against most other Gram-negative bacteria [115]. It influenced the transport of metal ions, the synthesis of peptidoglycan, the permeability of membranes, and the biosynthesis of enzymes in the cell; it could also inhibit biofilm formation in cariogenic *Streptococcus mutans* [136]. It is not used as an antibiotic in humans owing to its toxic effect [137]. Bacitracin A shows activity against rice pathogen *Pantoea ananatis* [114].

Bacitracin is able to inhibit the activity of subtilisin-like serine endopeptidases, porcine glutamyl and neutral aminopeptidases [138], and protein disulfide isomerase [139]. Bacitracin inhibits the activity of a highly glycosylated cell surface membrane serine aminopeptidase (porcine dipeptidyl peptidase-IV), which plays a role in tumor progression and glucose metabolism [140]. In addition, bacitracin showed dual specificity: as a metal-ion-independent RNase and as a magnesium-dependent DNase. It was able to degrade nucleic acids, being especially active against RNA molecules [141] (Figure 2). A disruption of peptidoglycan by prevention of the lipid II formation by bacitracin is described in [142] (Figure 3).

Six isolates of *B. licheniformis* from retail infant milk formulae (IMF1, IMF2, IMF5, IMF6, IMF 22, and IMF78) demonstrated higher antimicrobial potency than lichenicidin-producing strains. The result of further analyses identified a peptide of 1.422 kDa. This peptide showed high homology to the non-ribosomal peptides bacitracin and subpeptin, known to be produced by *Bacillus* spp. Strains IMF20, IMF66, IMF69, and IMF80 were also able to produce the two-peptide antibiotic lichenicidin [54].

Two antimicrobial peptides, subpeptin JM4-A and subpeptin JM4-B, with molecular masses of 1.42271 kDa and 1.42265 kDa, respectively, were produced by the soil isolate *Bacillus subtilis* JM4 [143,144].

*Bacillus licheniformis* strain EI-34-6 was isolated from the surfaces of the seaweed *P. palmata* and grown in an air-membrane surface (AMS) bioreactor. The cells produced antimicrobial compounds that were not previously produced when grown in shake flask cultures. The inhibitory compounds were active against *Staphylococcus aureus* strains MRSA9551 and MRSA14986 and vancomycin-resistant *Enterococcus* strains VRE788 and VRE1349. Glycerol and ferric iron were important for the production of antimicrobial compounds and red pigment, similar to pulcherrimin. The release of these secondary metabolites and bacitracin was not due to the onset of sporulation. The spent cell-free medium recovered from beneath the reactor membrane induced the production of antimicrobial compounds and red pigment in shake flask cultures. The antimicrobial compound was purified, and, on the basis of its chemical structure, it was determined to be bacitracin [145]. The supernatant produced by the bacteria was also capable of dispersing bacterial biofilms. The source of this activity was extracellular DNase (NucB), an enzyme that rapidly breaks up the biofilms of both Gram-positive and Gram-negative bacteria. The produced ribonuclease (barnase) may have an important role in dispersal efficacy [146].

#### 2.2.2. Cyclic Lipopeptides (Biosurfactants)

Biosurfactants include the amphiphilic compounds produced by microorganisms that show strong surface and emulsifying activity. These are microbial surfactants: chemically active compounds with an amphiphilic structure with hydrophilic (peptides or amino acids and polysaccharides) and hydrophobic (fatty acids) fragments. They are able to localize between liquids with different polarities, thereby reducing the surface and interfacial tension. They have a very low critical micelle concentration, no toxicity, high biodegradability and resistance to extreme conditions, such as high temperatures, extreme pH and high salinity [147]. Surfactants are used as cleaning agents, detergents, dispersants, moisturizers, and emulsifiers, and in the bioremediation of oil-contaminated sites [112]. Due to their antimicrobial and antiviral activities, they have been used to combat microbial and viral infections of plants [148]. The use of surfactants in the composition of antitumor drugs is effective [149,150,151]. Microbial surfactants have a number of advantages, such as biodegradability, operation in a wide range of pH and temperature, resistance to high concentrations of NaCl, higher selectivity and stability, and antibacterial and antifungal activities [152].

Several lipopeptide biosurfactants produced by *B. licheniformis* have antimicrobial activity [102,124,152,153] (Table 4). *B. licheniformis* is able to secrete biosurfactants (Table 1), such as lipopeptides, under various growth conditions: in the presence or absence of oxygen and under high-salinity and -temperature conditions [154]. They can be a useful tool to combat biofilm-forming bacteria. Lipopeptides are of particular interest because of their high surface tension and antibiotic potential [155].

A lipopeptide biosurfactant generally consists of a fatty acid chain and a peptide chain with several amino acids [155]. In lipopeptides a fatty acid residue is covalently linked to a peptide chain. This family includes surfactin, lichenysin, iturin and fengycin [156]. The relationship between the structure and functions of lipopeptides is expressed by varying degrees of antagonistic action depending on the pathogen, although in general they all cause the appearance of pores in cell membranes. *B. licheniformis* is capable of producing cyclic lipopeptides related to biosurfactants [157]. The results of an analysis of *B. licheniformis* isolated in seven different geographic areas showed that they differed in their lipopeptides content depending on the locality [158].

##### Surfactin Homologues

Surfactin is a well-characterized cyclic lipoprotein isolated from *Bacillus subtilis* and one of the most effective and powerful biosurfactant [159]. The surfactin family is a mixture of cyclic lipopeptides containing variants of a heptapeptide and a β-hydroxy fatty acid with chain length of 13–18 carbon atoms. A lactone bridge between the β-hydroxyl function of the acid and the carboxy-terminal function of the peptide confers a cyclic structure to the molecule [118]. When this lipopeptide interacts with Gram-positive bacteria, cell lysis is observed [160]. Surfactin is able to form pores in biological membranes and destabilize lipid packaging (Figure 2). Due to hydrophobic interactions, it binds to membranes and affects the ordering of hydrocarbon chains, which affects the thickness of the membrane [161]. Surfactin biosynthesis is catalyzed non-ribosomally via the action of a large multienzyme complex consisting of four modular building blocks, called surfactin synthetase [162].

*B. licheniformis* HSN221 produced nine variants of surfactin and lichenysin lipopeptides. The medium containing glucose, ammonium chloride, and yeast extract was especially suitable for the production of surfactin homologues [116,157]. The molecular masses of the two produced surfactin monomethyl esters and one lichenysin monomethyl esters detected via ESI-MS were 1.048, 1.049, and 1.063 kDa, respectively [163].

*B. licheniformis* BC98 inhibited the growth of phytopathogens, such as *Magnaporthe grisea*, *Curvularia lunata*, and *Rhizoctonia bataticola.* The active component had a molecular mass of 1.035 kDa. The active lipopeptide was identified as surfactin. The activity of the antagonistic lipopeptide was highly stable at extreme pH and temperatures; the antagonistic lipopeptide was also resistant to protease treatment. The result of the microscopic analysis of the effect of the antagonist on *M. grisea* revealed bulbous hyphae showing patchy and vacuolated cytoplasm. This lipopeptide was highly potent in its antagonistic activity; it completely inhibited the growth of *M. grisea* at a low concentration of 1 µg mL^–1^ [101].

The lipopeptides isolated from *B. licheniformis* supernatant [102] showed the highest structural analogy with the surfactin produced by *B. subtilis* [159]. The lipophilic part consisting of branched C_14_ or C_15_ hydroxy saturated fatty acids was linked to the hydrophilic peptide moiety, which contained seven amino acids (Glu, Asp, Val, three Leu, and Ile) via a lactone linkage. Antibiotic activity was demonstrated against Gram-negative bacteria (Pseudomonas aeruginosa and *Escherichia coli*), yeasts, and some fungi (*Trichoderma reesei* and *Penicillium oxalicum*). Two molecular weights, 1.022 and 1.036 kDa, were determined. The mass difference of 14 units characterized the lipopeptide as a mixture of closely related molecules varying in their fatty acid residues [102].

The lipopeptides produced by *B. licheniformis* MB01 were determined to be cyclic surfactin homologs with molecular weights 0.994, 1.008, 1.022, and 1.036 kDa. The lipopeptides demonstrated resistance to UV light and changes in pH and temperature. These surfactins were active against the Gram-positive and -negative bacteria (*Escherichia coli*, *Vibrio cholerae*, *Vibrio parahaemolyticus*, *Vibrio harveyi*, *Pseudomonas aeruginosa*, *Staphylococcus aureus*, and *Proteus species*) [117].

*B. licheniformis* V9T14 produced C_13_, C_14_ and C_15_ surfactin homologues, whose structures were confirmed by the product ion spectra of the sodiated molecules at *m*/*z* 1.030, 1.044 and 1.058 kDa, respectively [118]. The V9T14 biosurfactant was active against *Escherichia coli* CFT073 biofilm formation [119].

*B. licheniformis* B6 produced surfactin among other lipopeptides [127].

*B. licheniformis* ATCC 12713 produced surfactin that showed a strong antibacterial activity against *C. perfringens* and *Brachyspira hyodysenteriae*, the pathogens causing necrotic enteritis and swine dysentery, respectively. The major isoform of surfactin in *B. licheniformis* was found to be surfactin C [120]. The fermented products obtained from the same strain were able to inhibit the growth *of Staphylococcus aureus* in vitro; adding them to dietary feed could ameliorate *Clostridium perfringens* induced intestinal necrotic lesions in broilers [121,164]. This surfactin showed stronger bacterial killing activity against *C. perfringens* but not against the causative agent of swine dysentery, *Brachyspira hyodysenteriae*, unlike surfactin from *Bacillus subtilis* [120]. Furthermore, *B. licheniformis* ATCC 12713-derived surfactin exhibited anti-coccidial activity by inhibiting the life cycle of *Eimeria* species. This surfactin directly inhibited *E. tenella* oocyst growth in vivo, thereby preventing coccidiosis in broilers [165].

*B. licheniformis* 86 produced a mixture of lipopeptides with the major components ranging in size from 0.979 to 1.091 kDa and varying in increments of 14 Da. The most abundant components were of 1.021, 1.035 and 1.049 kDa in size. The data on the structure of this surfactant indicate its surfactin-like nature [122,166].

*B. licheniformis* F2.2 produced a non-lipopeptide type biosurfactant BL1193 together with lipopeptides, plipastatin, and surfactin in an amino acid-depleting medium. Plipastatin inhibited the growth of Gram-positive bacteria (*B. subtilis*), Gram-negative bacteria (*Pseudomonas aeruginosa* and *Escherichia coli*), and *Eumycetes* (*Aspergillus niger*, *Penicillium* sp., *Fusarium* sp., *and Cladosporium* sp.). Plipastatin and surfactin were abundantly produced in nutrient-rich medium. In addition, a non-lipopeptide-type biosurfactant BL1193 was produced upon the growth of the producer in a synthetic medium but not in a rich medium [103].

##### Lichenysins

A surface-active substance known as lichenysin is produced by *B. licheniformis* (Table 4) as a secondary metabolite, and its biosynthesis is catalyzed by non-ribosomal peptide synthetases. Its structure is similar to that of surfactin. Both compounds can be produced under aerobic or anaerobic conditions [167]. Lichenysin has a higher surfactant power and a much higher hemolytic activity than surfactin [124,168]. The main difference between lichenysin and surfactin is the presence of glutamine residue (Gln) at position 1 of the lichenysin peptide sequence in place of the glutamic acid (Glu) of surfactin, which results in changes in its physicochemical properties. Lichenysin is a better chelating agent toward Ca^2+^ than surfactin [168]. Some strains of *B. licheniformis* produced lichenysins and were mostly detected as sodium adducts at *m*/*z* 1.029 and with a size of 1.057 kDa [169].

*B. licheniformis* NBRC 104464 produced a cyclic lipopeptide different from surfactin, lichenysin with *m*/*z* 1.0295, 1.0435, and 1.0575. The association constant of this lichenysin with Ca^2+^ was four-fold higher than that of surfactin [123].

Both aerobically and anaerobically, *B. licheniformis* BAS50 produced lichenysin A, with the major components ranging in size from 1.006 to 1.034 kDa. Lichenysin A has an isoleucine as the C-terminal amino acid instead of the leucine of surfactin and lichenysin B and an asparagine residue instead of the aspartic acid residue of surfactin, lichenysin B, and lichenysin C. Glucose and sucrose, but not arabinose, fructose, or maltose, enabled the best surfactant production. The inhibitory activity was observed against *Acinetobacter calcoaceticus*, *Alcaligenes eutrophus*, *Bacillus cereus*, *Bacillus* sp. strain ATCC 39307, *Escherichia coli*, *Enterobacter* sp. strain 306, *Pseudomonas fluorescens*, *Pseudomonas proteofaciens*, *Staphylococcus aureus*. No growth inhibition by lichenysin A was detected for *B. licheniformis* BAS50, *B. subtilis*, or *Rhodococcus globerulus* [124].

The eight types of lichenysin commonly produced by *B. licheniformis* are lichenysin A, lichenysin B, lichenysin C, lichenysin D, lichenysin G, [Val7] lichenysin G, [Ile4] lichenysin G, and [Ile2,4] lichenysin G [124,170,171,172]. The lichenysin B-producing strain JF-2 was re-identified as *Bacillus mojavensis* strain JF-2 [124,173].

The lichenysin types differ due to the type and sequence of amino acids in the lactone ring [172].

In 1999, a series of nine lactone lipopeptide biosurfactants, representatives of the lichenisins group, was isolated from *B. licheniformis* strain IM 1307. According to the authors, they were at least 10 times more active than surfactins [171].

Later, nine lipopeptides (surfactins and lichenysins) produced by *B. licheniformis* HSN221 were identified via chromatography and mass spectrometry. By varying the composition of the nutrient medium, the strain produced either surfactins or lichenysins. The types of lipopeptides produced from natural substrates were the same, which contained lichenysin C13, lichenysin C14, and lichenysin C15. The lipopeptides produced from synthesized media were homologues of surfactin C13 and lichenysin C12. According to the structure of lichenysin A, the molecular masses of lichenysin C12, lichenysin C13, lichenysin C14, lichenysin C15 and lichenysin C16 are 0.992, 1.006, 1.020, 1.034, and 1.048 kDa, respectively [157].

Lichenysin showed toxic effects in pig ileum organoids and human epithelial CaCO_2_ cells. The concentration of lichenysin needed to reduce cell viability by 50% (IC_50_) was 16.6 µg/mL for Caco-2 human intestinal epithelial cells and 16.8 µg/mL for pig ileum organoids. For surfactin, the IC_50_ value was 23.5 µg/mL for Caco2 cells, whereas no toxicity was observed for the ileum organoids at the highest levels tested (>200 µg/mL). This indicated that lichenysin is more toxic to these cell types than surfactin [167].

*B. licheniformis* strain P40 produced an antibacterial cyclic peptide (BLS) that contains fatty acids, such as surfactin and lichenysin, but with a lower molecular weight of 0.8 kDa. It was resistant to temperatures up to 100 °C and pH ranging from 3 to 10, but lost its activity when treated with pronase E. However, it was resistant to papain, trypsin, proteinase K, and trichloroacetic acid. This peptide already demonstrated a wide-spectrum action, presenting bactericidal activity toward pathogenic and spoilage bacteria, such as *B. cereus*, *L. monocytogenes*, *E. carotovora*, and *Streptococcus* spp. However, *Staphylococcus aureus* and *Escherichia coli* were resistant to the action of this substance. The precipitation at a low saturation of ammonium sulfate and the elution at a void volume of gel filtration indicated that the BLS was secreted in the form of large aggregates [29,126].

##### Licheniformins

The physical properties and chemical structure of the licheniformin lipopeptide produced by *B. licheniformis* MS3 were studied [125]. The molecular weight of licheniformins is 1.438 kDa. This lipopeptide has a lactone ring consisting of four amino acid residues (Asp, Ser, Gly, and Tyr), which is additionally linked by an amide bond to the remaining amino acids (Gly, Ala, and Val). Hence, its peptide ring is not directly linked to the fatty acid moiety [125]. The structure of licheniformins is similar to that of the lipopeptide biosurfactant (kurstakin) produced by *Bacillus thuringiensis*, which shows antifungal activity against *Stachybotrys charatum* [174].

##### Fengycins

The fengycin family consists of a β-hydroxy fatty acid connected to the N-terminus of a decapeptide. The C-terminal residue of the peptide moiety is linked to the tyrosine residue at position 3, forming the branching point of the acyl-peptide and the eight-membered cyclic lactone. The length of the β-hydroxy fatty acid tail varies and links the amino group of its N-terminal amino acid Glu [118]. Fengycins exhibit antibacterial activity against both Gram-positive and Gram-negative microorganisms. In addition, these substances have been shown to be active against filamentous fungi [175,176,177]. Being a surfactant, fengycins interact with biological membranes and form pores in them, which leads to a change in the permeability of the membrane [178] (Figure 2). Its action is associated with a modification of the alignment of the phospholipid acyl chain and a global decrease in the cooperativity of the lipid–lipid and lipid–fengycin interaction in the bilayer membrane [179]. This effect may be related to the ability of fengycins to change the hydrophobicity of the bacterial surface, influence the development of biofilms and flagella, and prevent the attachment of bacterial cells to various surfaces, including plastic, glass, and tissues [117,180,181].

*B. licheniformis* B6 produced lipopeptides (LPs) that demonstrated antibacterial activity against clinical pathogenic strains *Staphylococcus aureus*, *Escherichia coli*, and *Klebsiella* sp. In the presence of LPs, biofilm structures were destabilized, turning these strains into weak biofilm-formers. Kurstakin and iturin were identified via MALDI TOF. The mass spectra revealed mass peaks assigned to fengycins and bacitracins ranging from *m*/*z* 0.850 to *m*/*z* 1.200 kDa; and assigned to the isoforms of kurstakins, surfactins, and iturins ranging from *m*/*z* 1.300 and *m*/*z* 1.650 kDa. Surfactin was detected, rather than lichenysin, which is the lipopeptide expected to be produced by *B. licheniformis* species. Signals of bacitracin and fengycins were also found, the latter with a higher number of homologues and relative intensity than the other lipopeptides. These results showed that the lipopeptides synthesized by *B. licheniformis* B6 have both antibacterial and antibiofilm activity against pathogenic bacteria of health importance [127].

The lipopeptide biosurfactants produced by the *B. licheniformis* strain V9T14 showed an anti-adhesion activity against the biofilm formation of human pathogenic bacterial strains. The presence of two main fengycin isoforms was found, with protonated molecules at *m*/*z* 1.478 and 1.506 kDa, corresponding to C_17_ fengycin A and C_17_ fengycin B, respectively. Other homologues (C_14_ to C_16_) were revealed and confirmed as belonging to fengycin A or B [118]. The biosurfactants produced by V9T14 inhibited *E.coli* and those produced by V19T21 inhibited *S. aureus* biofilm formation [119] Moreover, the V9T14 biosurfactant was able to increase the biofilm eradication efficacy of different antibiotics against an uropathogenic *Escherichia coli* strain [182].

#### 2.2.3. Other Lipopeptides

*B. licheniformis* strain M104, when grown on whey, produced a lipopeptide biosurfactant with activity against Gram-positive bacteria (*Bacillus subtilis*, *Bacillus thuringiensis*, *Bacillus cereus*, and *Staphylococcus aureus*) and Gram- negative bacteria (*Pseudomonas aeruginosa*, *Escherichia coli*, *Salmonella typhimurium*, and *Proteous vulgaris*), as well as yeast (*Candida albicans*). *Listeria monocytogenes* and *Klebsiella pneumoniae* were resistant to the action of this biosurfactant [153]. Its chemical structure was not established.

*B. licheniformis* 603, isolated from a mixture of drilling fluid and subsurface thermal water, produced a cyclic lipopeptide with growth-inhibiting activity against *Corynebacterium variabilis* and *Acinetobacter* sp. Additionally, this lipopeptide prevented the adhesion of bacterial cells to a glass surface. This compound was a heptapeptide containing L-Asp, L-Leu, L-Leu, L-Val, L-Val, L-Glu, and L-Leu, with N-acylated to the N-terminal amino acid (L-Asp) by a 3-hydroxy fatty acid, the 3-OH group of which was esterified by the C-terminal amino acid, L-Leu [128].

CB-1 is a unique chitin-binding antifungal including peptides and fatty acids. It is an aggregation product of four peptides of 1.035, 1.504, 4.018, and 5.024 kDa. According to the results of gel filtration column chromatography, the molecular mass was estimated as 42 kDa. It showed inhibitory activity against some phytopathogenic fungi, including *Pyricularia oryzae* and *Rhizoctonia solani*, and less activity against bacteria and yeast [104].

A lipopeptide surfactant from the marine sponge-associated *Bacillus licheniformis* NIOT-AMKV06 showed antimicrobial activity against life-threatening clinical pathogens, such as *Enterococcus faecalis*, *Bacillus subtilis*, *Salmonella typhi*, *Vibrio cholera*, and *Klebsiella pneumoniae* and some other bacteria [129].

Thus, the surfactants synthesized by *B. licheniformis* have the potential to inhibit the growth and biofilm formation of human and animal pathogenic bacteria, mainly Gram-positive ones, such as *Staphylococcus aureus*, *Listeria monocytogenes*, and *B. cereus*, and some Gram-negative bacteria, including *Escherichia coli*, *Salmonella Typhimurium*, and *Aeromonas* sp. [29,31,177,183,184]. However, due to their toxicity towards animal and human cells, their application in medical and veterinary practice is limited to topical usage and in the form of disinfectants.

Many antimicrobial and antifungal peptides and proteins produced by *B. licheniformis* are resistant to the action of proteinases. Perhaps the stability of these proteins can be explained by the presence of a cyclic peptide structure in these bacteriocins containing unusual amino acids [185].

### 2.3. Exopolysaccharides

Exopolysaccharides (EPSs) are high molecular weight compounds composed of repeated units of sugar moieties, attached to a carrier lipid, and associated with proteins, lipids, organic and inorganic compounds (acetate, glycerol, pyruvate, sulfate, carbo xylate, succinate, and phosphates), metal ions, and DNA [186]. In some cases, EPSs demonstrated antimicrobial activity against bacterial pathogens (Table 5), both Gram-positive and Gram-negative. Their antagonistic action was revealed in relation to bacteria, viruses, and fungi. EPSs also inhibit the formation of biofilms by pathogenic bacteria and prevent their colonization on various surfaces [187].

*B. licheniformis* can synthesize EPSs with various biological activities (Table 1), including those with antibacterial and antioxidant effects [189]. A typical example is levan (fructan), which includes fructose polymers linked with β-2,6-fructofuranosidic bonds. Levan is synthesized by an enzyme, levansucrase. It has antioxidant and antibacterial activities against *Staphylococcus aureus*, *E. coli*, and *Pseudomonas aeruginosa* [188]. *B. licheniformis* RN and *B. licheniformis* SVD1 produced levans that showed potential as substances with antibacterial, antibiofilm, antiviral, and anticarcinogenic effects [192,193].

*B. licheniformis* 24 produced EPSs consisting of galactose, glucose, and mannose and showed antioxidant activity. Additionally, these EPSs possessed antibacterial activity against *Vibrio cholera* [189].

*B. licheniformis* Dahb1 produced EPS with antioxidant and antibiofilm/antibacterial activity against Gram-negative (*Pseudomonas aeruginosa* and *Proteus vulgaris*) and Gram-positive species (*Bacillus subtilis* and *Bacillus pumilus*) as well as the fungus *Candida albicans*. The contents of carbohydrates, proteins, and uronic acid in EPSs were 680.43, 386.15, and 56.72/mg, respectively. The results of the hemolytic assay showed low cytotoxicity of this EPS at 5 mg/mL [106].

*B. licheniformis* T14 produced EPS-T14 (molecular weight of 1000 kDa) with antibiofilm activity. It contained fructose and fucose as the major monosaccharides. EPS-T14 reduced the biofilm formation of both Gram-negative and Gram-positive bacteria (multiresistant clinical strains of *Escherichia coli*, *Klebsiella pneumoniae*, *Pseudomonas aeruginosa*, and *Staphylococcus aureus*) [190].

The exopolysaccharide (1800 kDa) purified from the culture supernatant of sponge-associated *B. licheniformis* was able to inhibit biofilm formation of *E. coli* and *Pseudomonas fluorescens* but not able to reduce the growth of these bacteria. This EPS was composed of a-D-galactopyranosyl-(1→2)-glycerol-phosphate monomeric units [191].

## 3. Antimicrobial Substances of *B. licheniformis* Active against Mycobacteria

Most of the antibacterial components produced by different strains of *B. licheniformis* are active only against Gram-positive microorganisms. Some are also active against Gram-negative microorganisms. Few substances have been reported to be active against mycobacteria (Table 6), whose cell wall differs from that of Gram-positive and Gram-negative bacteria and effectively functions as a permeability barrier [194].

*Mycobacterium tuberculosis* causes a respiratory tract infection known as tuberculosis. On average, 10 million people worldwide are infected with this disease each year, and the mortality rate is between 11 and 15%. Cases of multidrug and extensive resistance in *M. tuberculosis* have limited the efforts to control its spread, especially in developing countries [195]. In addition to drug resistance, the causative agent of tuberculosis (*Mycobacterium tuberculosis* (Mtb)) is able to asymptomatically persist in the host organism for many years, causing latent forms of tuberculosis. In this dormant state, Mtb cells are also resistant to known antibiotics [2,3]. Due to its unique metabolic plasticity, the mycobacterium survives under the stressful conditions of the host organism and under antibiotic therapy. In these cases, the mycobacteria can gradually move into a state of reduced metabolic activity, a dormancy associated with the ineffective treatment of latent tuberculosis infection [196].

Infection with COVID-19 often results in the transition of latent tuberculosis to an active form, which, in a significant percentage of cases, is drug-resistant [197]. According to the WHO, every one in four inhabitants of the planet is an asymptomatic carrier of tuberculosis; thus, a permanent reservoir exists of tuberculosis infection, from which a pandemic can develop at any moment. The lockdowns and restrictions imposed during COVID-19 may lead to an additional 1.4 million TB deaths between 2020 and 2025, according to the WHO [195]. Thus, the search for new substances capable of killing mycobacteria is an important task in medical microbiology and chemistry.

The unique structure of the mycobacterial cell wall and the characteristic slow growth of *M. tuberculosis* may presumably interfere with the action of lantibiotics (Figure 2). Lantibiotics can bind to the lipid II of mycobacteria, making them candidates as anti-tuberculosis drugs. The lipid II structure of mycobacteria is different from that of other bacteria, including modifications of both N-acetylmuramic acid (MurNAc) and the side chain of the peptide [198].

The nisin produced by lactococci showed activity against mycobacteria *M. smegmatis* and *M. bovis*, with intracellular ATP leakage and proton motive force dissipation. Nisin and lacticin were also active against clinical isolates of mycobacteria in vitro, including *M. tuberculosis* [199,200]. *B. licheniformis* MCC 2512^T^ produced a natural variant of nisin, subtilin [90,91], which was active against *M. tuberculosis* [201] (Figure 2).

In 1946, it was demonstrated that *B. licheniformis* produced several antibacterial substances that inhibit the growth of mycobacteria, including the causative agent of tuberculosis, *M. tuberculosis* [202]. One of these substances was named licheniformin. Later, this compound was found, however, to be toxic, causing damage to the kidneys after prolonged administration [65]. In the following study, it was revealed that *B. licheniformis* produced three similar components, designated as licheniformins A, B, and C. These peptides have similar molecular weights and amino acid compositions, possessing both antibacterial activity and toxicity, although to somewhat different degrees. All three peptides have similar molecular weight (3.8–4.8 kDa), optical rotation, and elemental compositions. Purified licheniformin C was less active against mycobacteria than the original crude preparation and caused more pronounced kidney damage. Licheniformin B was slightly more active in vitro than the parent substance but also caused extensive renal damage. Licheniformin A was much less toxic than either of the other fractions but still caused some kidney damage and was less effective than streptomycin in controlling tuberculosis in mice. Licheniformins A and B were more active against *Mycobacterium phlei* than licheniformin C and less toxic to mice than licheniformin C [37].

Different species of laboratory animals are not equally susceptible to the nephrotoxic action of licheniformin A5. Compared with mice, rabbits are resistant, and rats are relatively sensitive [203]. However, the nephrotoxicity suspended further work with these compounds despite their high effectivity as anti-TB substances in vitro. In addition to inhibiting the growth of mycobacteria, licheniformins showed efficacy against *Staphylococcus aureus* and *Escherichia coli* [65].

Bacitracin, at concentrations of 6.5–13.0 µg/mL, inhibited the growth of *Mycobacterium smegmatis*. For the inhibition of *M. tuberculosis* BCG, the concentration of bacitracin was 10 times higher. The main target of the action of bacitracin on mycobacteria is presumably the membrane system (Figure 2). Bacitracin caused marked alterations in mycobacterial membranous structures. Bacitracin is highly bactericidal to mycobacteria during the middle or late exponential growth phase [204].The structure of bacitracin is shown in Figure 4 and Figure 5.

The strain *B. licheniformis var. mesentericus* produced proticin, which is especially active against a number of Gram-positive and Gram-negative bacteria, including mycobacteria (*Mycobacterium tuberculosis*). The median lethal dose of proticin for mice was >150 mg/kg intravenously and 1000 mg/kg subcutaneously [38]. Proticin is a phosphorus-containing, strongly unsaturated, amorphous compound with a conjugated triene and a molecular weight 0.560 kDa. On the basis of this derivative and several degradation products, the molecular formula of proticin was found to be C_31_H_44_O_7_PNa. The functional groups of proticin include one OH capable of acetylation, one lactone group, and one monoester of phosphoric acid as enol ester. Proticin contains a conjugated triene [205].

Three hydrophobic peptides (amoebicins d13-A, d13-B, and d13-C) and peptide A12-C from B. licheniformis are able to inhibit Mycobacterium phlei [46,108].

Activity against *M. smegmatis* and *M. phlei* is known for lichenicidin VK21 [52]. AlpaFold prediction of lichenicidin VK21 structure is shown in Figure 6 below.

According to our unpublished observations, a laboratory strain of *B. licheniformis* LBSM secreted anti-*M. tuberculosis* 14 kDa substance(s), which inhibited the growth of multiple cells and destroyed dormant *M. tuberculosis* forms. This substance was resistant to proteinase action.

Although these bacteriocins have potential, in vivo studies are still required, and an appropriate delivery system still needs to be developed to reach the *M. tuberculosis* residing within tissues. For example, in the context of *M. tuberculosis*-infected macrophages in the distal lung, promising results have been reported regarding the in vivo efficacy of class IIa bacteriocins complexed with phosphatidylcholine–cardiolipin liposomes. As a complex with liposomes, bacteriocins inhibited the intracellular growth of *M. tuberculosis* to prolong the survival of mice in an acute TB model [206].

## 4. Prospects for Using Natural Substance in the Treatment of Tuberculosis

Natural producers of antimicrobial compounds are attractive starting points for finding new and better anti-tuberculosis drugs because they are rich in chemical diversity and have strong antimicrobial activity [207]. Traditionally, natural products have been used as the prototype of the various drugs that are currently actively used in medicine. These include pyrans, flavones, chalcones, coumarins, pyrimidones, and oxzolidines, which are used as anti-cancer, anti-inflammatory, antimicrobial, antiviral, and anti-tuberculosis medicines [207].

The above-mentioned examples demonstrated that the bacteriocins of different bacteria exhibit stronger in vitro antimycobacterial activity than equal concentrations of rifampicin, which is a widely used anti-TB antibiotic. They can be considered alternatives for the development of methods to combat the antibiotic-resistant strains of mycobacteria that cause tuberculosis.

Antimicrobial peptides are capable of disrupting the normal function of the mycobacterial cell wall through various approaches and then interacting with different intracellular targets (including nucleic acids and enzymes) [208]. Importantly, the likelihood of developing resistance to antimicrobial natural peptides is rather low. This is first due to their non-specific mode of action and, second, because the same molecule has different mechanisms of destruction. In addition, mutations that make bacteria resistant to bacteriocins are energy-intensive and harmful [209]. Usually, these peptides have a positive charge and can interact with a negatively charged mycobacterial cell wall [210]. As a result of this interaction, peptides enter the cytoplasm, where they can interact with intracellular targets. Due to their amphipathic nature, antimicrobial peptides can be active in both aqueous and lipid environments [211]. The interaction of bacteriocins with the mammalian cell membrane is weaker than that with the bacterial membrane. This is due to the different compositions and structures of lipids. Mammalian phospholipids are mostly zwitterionic, resulting in a neutral charge, whereas bacterial membranes have a negatively charged outer surface [212]. In mammalian membranes, zwitterionic phospholipids are found in the outer leaflet, whereas negatively charged phospholipids are found closer to the cytoplasm in the inner leaflet. The interaction of antimicrobial peptides and mammalian cell membranes is possible because hydrophobic contacts, which are weaker than the electrostatic interactions between bacteriocins and bacterial membranes. The presence of cholesterol, which stabilizes the phospholipid bilayer of mammalian membranes, reduces the activity of antimicrobial peptides [213]. Thus, because of structural differences between mammalian and bacterial membranes, peptides selectively act on bacterial cells rather than mammalian cells, indicating their suitability as therapeutic agents against pathogenic bacteria [214]. In summary, natural bacteriocins possess evident advantages in comparison with traditional antibiotics.

Many antibacterial peptides are resistant to proteases, which makes them suitable for intravenous or per os administration. Nevertheless, the medical application of some bacteriocins with promising in vitro antimycobacterial activity is limited by their toxicity toward animals and humans.

The recent technological advances have allowed the production of new antimicrobials through the structural modification of natural peptides to overcome resistance to antibiotics [215].

To enhance the antimycobacterial activity of natural bacteriocins, as well as to reduce their toxicity, biotechnological approaches are used. Through the use of these approaches, the activity of the biotechnological derivatives of nisin were enhanced against mycobacteria compared with that of the prototypical substance [216]. We might expect that a similar approach could be used in medical studies and eventually in the application of the licheniformins (see above) that were efficient in vitro which were discovered in the last century. The generation of mycobacterial species-specific bacteriocins would be an exciting step forward in the development of novel antimycobacterial drugs.

Because many bacteriocins are synthesized on ribosomes, genes that encode the structural (though as yet inactive) peptide, bacteriocins are probably more convenient for bioengineering than classical antibiotics. The latter are usually generated from small building blocks through multienzyme complexes and are not ribosomal in nature. Various strategies have been developed to modify the properties of natural bacteriocins [217,218].

Natural biosurfactants with antimicrobial, antibiofilm, and antiviral properties may be applied for the production of disinfectants, handwashing, and cleaning products that are active against mycobacterial contaminations as well. They exhibit higher biodegradability, lower toxicity, and better environmental compatibility than synthetic surfactants [219].

The synergistic effect of natural bacteriocins and traditional antibiotics may allow more successful treatment of patients with fewer side effects [220].

## 5. Conclusions

*B. licheniformis* is an organism in the bacterial world that has an effective bacterial antagonism system based on the production (secretion) of antimicrobial and antifungal substances with different structures targeting many bacterial and fungal representatives, including those that are pathogenic to animal and humans. These substances are represented by peptides, proteins, lipopeptides, and polysaccharides. Peptides and proteins are classified on the mode of their synthesis by the producer strain: bacteriocins, peptides or proteins of different structural composition and molecular mass 1.5–20 kDa synthesized by bacteria ribosomally; non-ribosomally synthesized peptides and cyclic lipopeptides; and exopolysaccharides. Based on thermostability, size and chemical moieties, bacteriocins are classified into several groups. Among them, peptides that are active only against Gram-positive microorganisms and active against both Gram-positive and Gram-negative microorganisms can be distinguished. Peptides with antifungal and amoebolytic activity were also identified. The main mechanisms of their lytic activity include interaction with membranes of target cells resulting in pore-forming and inhibition of cell wall biosynthesis. DNase and RNase activity for some bacteriocines are also postulated.

Non-ribosomal peptides are synthesized by special non-ribosomal multimodular peptide synthetases and contain unnatural amino acids. Bacitracin (a mixture of cyclic peptides of ca. 1.5 kDa) is a widely known antibiotic of this group which is active in respect to Gram-positive microorganisms and is actively used in medicine and veterinary medicine. However, in contrast to bacteriocines, bacitracin blocks cell wall formation. Cyclic lipopeptides (biosurfactants) are non-ribosomal peptides which are chemically active compounds of amphiphilic structure with hydrophilic (peptides or amino acids, and polysaccharides) and hydrophobic (fatty acids) fragments. They are active against the Gram- positive and Gram-negative bacteria and phytopathogens. Harmful effect of lipopeptides arises due to their ability to form pores in biological membranes and destabilize lipid packaging. They can be a useful tool to combat biofilm-forming bacteria.

Secreted polysaccharides are high molecular weight compounds and composed of repeated units of sugar moieties attached to a carrier lipid. They demonstrate antimicrobial activity against bacterial pathogens, both Gram-positive and Gram-negative. Their antagonistic action was revealed in relation to bacteria, viruses, and fungi. EPSs also inhibit the formation of biofilms by pathogenic bacteria and prevent their colonization on various surfaces [172].

*B. licheniformis* possess remarkable plasticity as a secretion vehicle. Thus, the repertoire of substances secreted by *B. licheniformis* depends on the growth medium composition for the particular strain and differs for different strains, which allows to modulate the type of secreted substances depending on the particular need.

Some of them are currently in use in medical and veterinary practice (e.g., bacitracin or surfactins). However, many of them have been studied in vitro only and are awaiting in vivo experiments.

A number of peptides and proteins secreted by *B. licheniformis* revealed strong antimycobacterial activity. However, medical application of some substances with promising in vitro antimycobacterial activity (like licheniformins or bacitracin) is limited by their toxicity in animals and humans.

Studying the relationship between the peptide structure, function, toxicity, and molecular mechanism of action can provide a more complete understanding of peptides and the development of strategies required to modify them. This information will be useful in the development of new molecules with desired properties by applying modern biotechnological approaches.

On the other hand, particular strains of *B. licheniformis* could be used as a natural vehicle for antibiotic substance in the form of true probiotic cultures strains to combat various types of pathogens, including mycobacteria. Moreover, current technologies allow to construct *B. licheniformis* strains producing multiply antibacterial peptides or proteins or their combinations directed against particular pathogen. In this case, antibacterials would be continuously produced for a long time until extinction of the producer strain from the intestinal tract. In addition, intestinal localization of multiplying *B. licheniformis* will protect secreted active substances from aggressive action of the stomach environment. However, more studies are needed for the exploration and development of prospective capabilities of *B. licheniformis* to synthesize and produce a bouquet of different antibacterial compounds for application in medicine and veterinary fields.

## Figures and Tables

**Figure 1 pharmaceutics-15-01893-f001:**
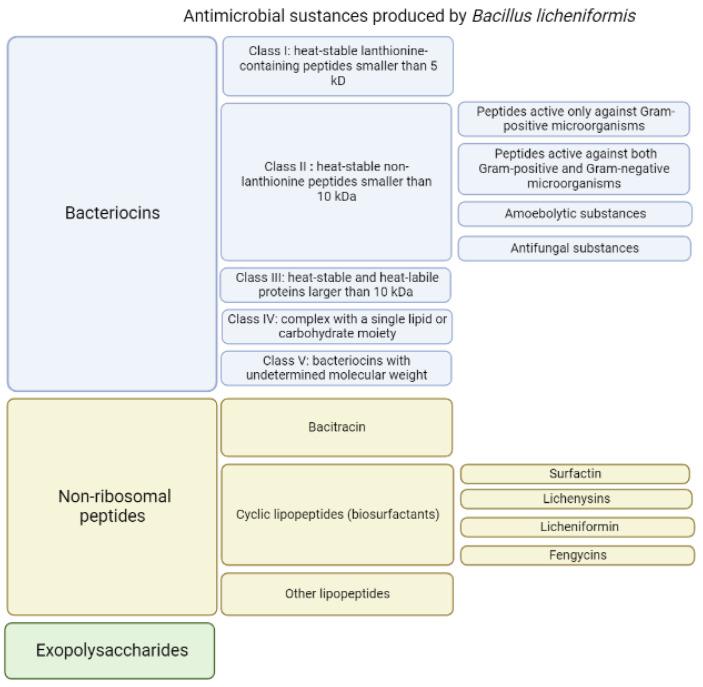
Antibacterial substances produced by *Bacillus licheniformis*.

**Figure 2 pharmaceutics-15-01893-f002:**
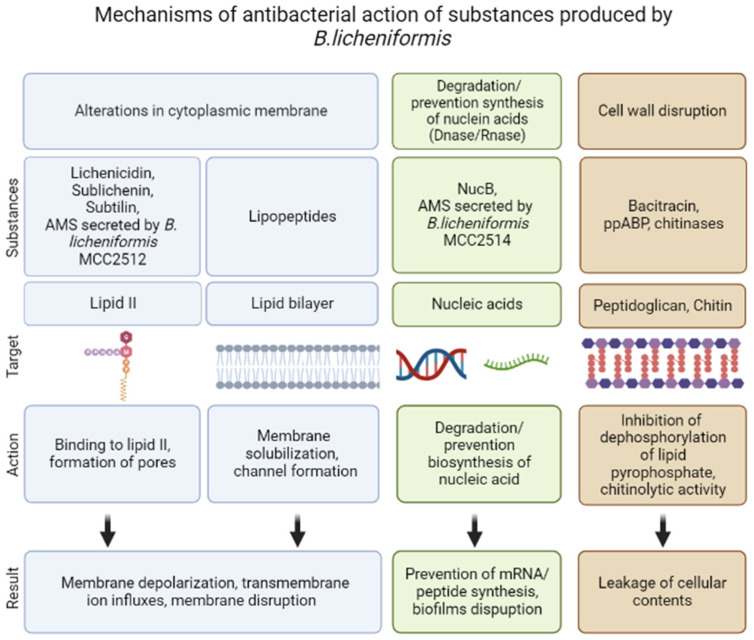
Mechanisms of antibacterial action of substances produced by *Bacillus licheniformis*.

**Figure 3 pharmaceutics-15-01893-f003:**
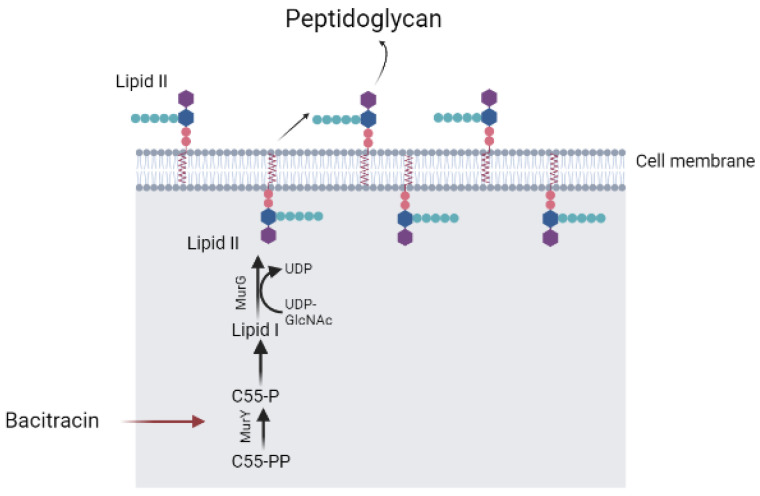
Disruption of peptidoglycan by prevention of the lipid II formation by bacitracin.

**Figure 4 pharmaceutics-15-01893-f004:**
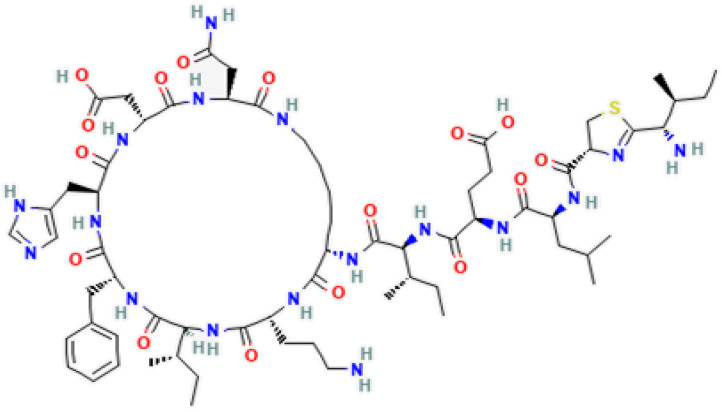
Chemical structure of bacitracin A (PubChem).

**Figure 5 pharmaceutics-15-01893-f005:**
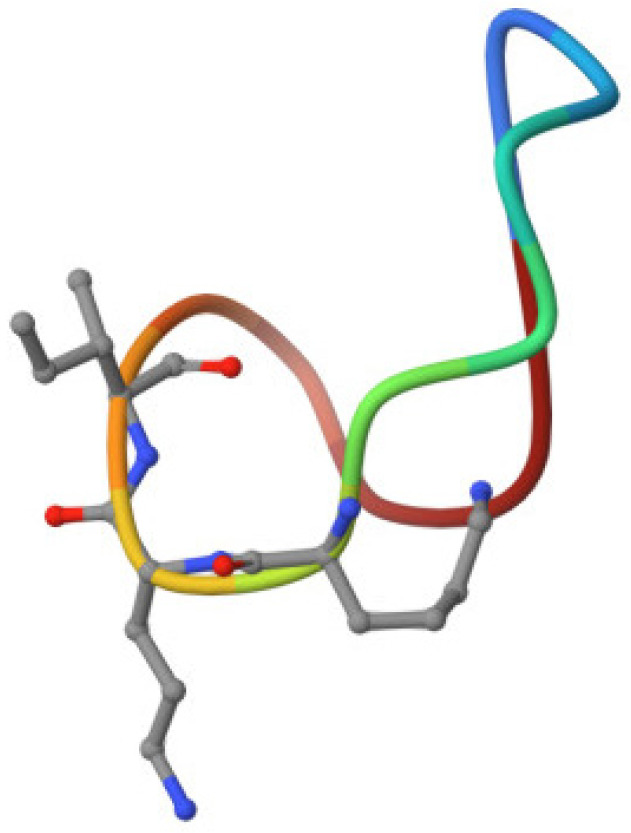
Tertiary structure of bacitracin (predicted by AlphaFold).

**Figure 6 pharmaceutics-15-01893-f006:**
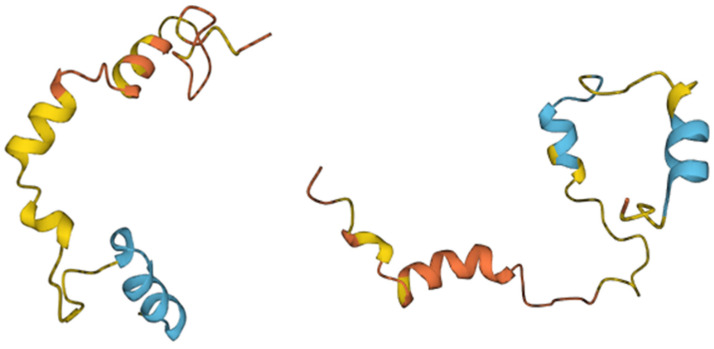
AlphaFold predicted structures of lichenicidin subunits.

**Table 2 pharmaceutics-15-01893-t002:** Antifungal substances produced by *Bacillus licheniformis*.

Substance(s) Specific/Unspecific Name	Producing Strain	Molecular Mass	Activity Assay	Reference
Bacteriocin-like peptides	*Bacillus licheniformis* ZJU12	3 kDa	*Xanthomonas oryzae pv. oryzae* Zhe 173 *Alternaria brassicae* (*cabbage isolate*) *Fusarium oxysporum* (*cotton isolate*) *Guignardia* sp. (*shihu isolate*) *Pyricularia grisea* (*rice isolate*) *Rhizoctonia solani* (*rice isolate*)	[33]
Antibiotics culture filtrate	*Bacillus licheniformis* strain MGrP1	ND	*Colletotrichum lindemuthianum* *C. kahawae* *Fusarium oxysporum* f.sp. *phaseoli* *Alternaria solani*	[96]
Fungicin M-4	*Bacillus licheniformis* M-4	3.6 kDa	*Microsporum canis* CECT 2797 *Mucor mucedo* CECT 2653 *Mucor plumbeus* CCM F 443 *Sporothrix schenckii* CECT 2799	[97]
Amoebicins M4—a, b, c	*Bacillus licheniformis* M-4	3–3.2 kDa	*Aspergillus niger* CECT 2089 *Candida albicans* CECT 1394 *Cryptococcus neoformans* CECT 1075 *Microsporum canis* CECT 2797 *Mucor mucedo* CECT 2653 *Mucor plumbeus* CCM F443 *Penicillium* sp. *Rhizopus oryzae* CECT 2340 *Saccharomyces cerevisiae* *Sporothrix schenckii* CECT 2799 *Trychophyton mentagrophytes* CECT 2793	[98]
Peptide A12-C	*B. licheniformis* A12	0.77 kDa	*Microsporum canis* CECT 2797 *Mucor mucedo* CECT 2653 *M. plumbeus* CCM F 443 *Sporothrix schenckii* CECT 2799 *Trychophyton mentagrophytes* CECT 2793	[46]
QSM (ComX pheromone)	*B. licheniformis* NCIMB 8874	ND	*A. flavus* NRL 3375, ESP 15	[99]
Bacteriocin MKU3	*B. licheniformis* MKU3	1.5 kDa	*Candida albicans* MTCC 183 *Aspergillus niger* MKU1 *Aspergillus fischeri* FXN1 *Aspergillus fumigatus* MKU3	[34]
Antifungal protein	*B. licheniformis* HS10	55 kDa	*Phytophthora capsici* *Botrytis cinerea* *Sclerotinia sclerotiorum* *Bipolaris maydis* *Fusarium graminearum* *Bipolaris sorokinianum* *Gaeumannomyces graminis*	[100]
Chitinase	*B. licheniformis* MY75 (also produced by Mb-2,TP–1, S213, SSCL-10, B307 strains)	55 (67,68/62/60,65,66) kDa	*G. saubinetii* *A. niger* (*Phoma medicaginis*)	[69,70,71,72,73,74,75]
Antifungal Protein F2	*B. licheniformis* BS-3	31 kDa	*Aspergillus niger* *Magnaporthe oryzae* *Rhizoctonia solani* *Fusarium oxysporum* (schl.)f.sp. *momordicae*	[76]
Ieodoglucomides A and B	*B. licheniformis* 09IDYM23	ND	*C. albicans* *A. niger*	[80]
Ieodoglucomide C and ieodoglycolipid	*B. licheniformis* 09IDYM23	ND	*C. albicans* *A. niger* *R. solani* *C. acutatum* *B. cenerea*	[81]
F4, F5 and F6	*B. licheniformis* BFP011	Less than 45 kDa	*C. capsici*	[79]
Bacteriocin	*B. licheniformis* HJ2020 MT192715.1	ND	*Candida albicans*	[85]
Surfactin	*B. licheniformis* BC98	1.035 kDa	*Sclerotium sclerotinii* *Phomopsis phyllanthi* *Rhizoctonia bataticola* *Aspergillus niger* N 573 *Curvularia lunata* *Magnaporthe grisea* *Helminthosporium* sp. *Chaetomium* sp. *Fusarium verticillioides* *Pestaliopsis magnifera* *Gleosporium magnefera*	[101]
Lipopeptides	*B. licheniformis* (soil isolate)	1.022 and 1.036 kDa	*Candida utilis* *C. tropicalis* *Trichosporon cutaneum* *Saccharomyces cerevisiae* *Trichoderma reesei* *Penicillium oxalicum*	[102]
BL1193 (non-lipopeptide-type biosurfactant together with lipopeptides, plipastatin, and surfactin)	*B. licheniformis* F2.2	1.193 kDa	*Aspergillus niger* *Penicillium* sp. *Fusarium* sp. *Cladosporium* sp. (inhibited by plipastatin)	[103]
CB-1	*Bacillus licheniformis*	42 kDa	*Pyriculariz oryzae* MAFF 101002 *Rhizoctonia solani* CF-1 *Corticium rolfsii* MAFF 712043 *Tyromyces palustris* MAFF 420001 *Botrytis cinerca* MAFF 712057 *Coriolus versicolor* CF-2 *Fusarium oxysporum* NFRI 1011 *Saccharomyces cerevisiae* Y02587	[104]
Biosurfactant	*Bacillus licheniformis* M104	ND	*Candida albicans* ATCC 70014	[105]
Bl-EPS	*B. licheniformis* Dahb1	ND	*C. albicans*	[106]
Peptide A12-A, A12-B	*Bacillus licheniformis* A12	1.43–1.6 kDa	*Candida albicans* CECT 1394 *Cryptococcus neoformans CECT* *Saccharomyces heterogenicus* *Aspergillus niger* CECT 2089 *Microsporum canis CECT 2797* *Mucor mucedo CECI 2653* *Mucorplumbeus CCM F443* *Sporothrix schenchii CECT 2799* *Ttychophyton mentagrophytes CECT 2793* *B. megaterium* *Cotynebactenum glutamicum CECT 78* *Sarcina* sp.	[107]
BL-DZ1 (BL00275)	*B. licheniformis* strain D1	14 kDa	*Candida albicans* BH	[67]

**Table 3 pharmaceutics-15-01893-t003:** Amoebolytic substances produced by *Bacillus licheniformis*.

Substance(s) Specific/Unspecific Name	Producing Strain	Molecular Mass	Activity Assay	Reference
Peptide A12-A и A12-B	*Bacillus licheniformis* A12	1.43–1.6 kDa	*N. fowlen* S-3 (=ATCC 30809) *N. fowlen* HB-1 (=ATCC 30174) *N. lovaniensis* Aq/9/1/45D *N. gruberi* CCAP 1516/le *B. megaterium* *Cotynebactenum glutamicum* CECT 78 *Sarcina* sp. *B. megaterium* *Cotynebactenum glutamicum* CECT 78 *Sarcina* sp.	[107]
Amoebicins M4—a, b, c	*Bacillus licheniformis* M-4	3–3.2 kDa	*Acanthamoeba* sp. Gr-1 *Naegleria fowleri* S-3 (=ATCC 30809) *N. fowleri* HB-1 (=ATCC 30174) *Naegleria lovaniensis* Aq/9/1/45D *Naegleria gruberi* CCAP 1516/le	[98]
Amoebicins d13-A, d13-B and d13-C	*B. licheniformis strain* D-13	1.87 kDa	*Acanthamoeba* sp. strain Gr-1 *Naegleria lovaniensis* Aq/9/1/45D *Naegleria gruberi* CCAP 1516/le *N. fowleri* S-3 (=ATCC 30809) *N. fowleri* HB-1 (=ATCC 30174) *Alcaligenes facecalis* *B. licheniformis M-4*, *A12* *Bacillus megaterium* *Corynebacterium glutamicum* CECT 78 *Enterococcus faecalis* S-13, S-14, S-48, S-86 *Micrococcus luteus* *Mycobacterium phlei* *Pseudomonas reptilovora* N5	[108]

**Table 4 pharmaceutics-15-01893-t004:** Antimicrobial non-ribosomal biosynthesized peptides produced by *Bacillus licheniformis*.

Substance(s) Specific/Unspecific Name	Producing Strain	Molecular Mass	Activity Assay	Reference
2.2.1. Bacitracin				
Bacitracin/Ayfivin	*Bacillus licheniformis* strain EI-34-6, HN-5	1.42 kDa	*M. tuberculosis* *M. smegmatis* *Actinomyces israeli* *Pantoea ananatis* *gram-positive cocci* *staphylococci* *streptococci* *corynebacteria* *anaerobic cocci* *clostridia*	[112,113,114,115]
			*Treponema pallidum* *T. vincenti* *neisseria* *gonococci* *meningococci*	
Antimicrobial compound (a variant of subpeptin and bacitracin)	*B. licheniformis* IMF1, IMF2, IMF5, IMF6, IMF22 and IMF78	1.42 kDa	*L. lactis* HP *L. bulgaricus* LMG 6901 *S. aureus* ST528 *S. agalactiae* ATCC 13813 *L. innocua* FH2333 *L. monocytogenes* LO28	[54]
**2.2.2. Cyclic lipopeptides (biosurfactants)**				
**2.2.2.1. Surfactin** **homologues**				
Surfactin and lichenysin isoforms	*Bacillus licheniformis* HSN 221	1.048–1.063 kDa	ND	[116]
Surfactin	*B. licheniformis* BC98	1.035 kDa	*Sclerotium sclerotinii* *Phomopsis phyllanthi* *Rhizoctonia bataticola* *Aspergillus niger* N 573 *Curvularia lunata* *Magnaporthe grisea* *Helminthosporium* sp. *Chaetomium* sp. *Fusarium verticillioides* *Pestaliopsis magnifera* *Gleosporium magnefera*	[101]
Lipopeptides	*B. licheniformis* (soil isolate)	1.022 and 1.036 kDa	*Pseudomonas aeruginosa* *Escherichia coli*	[102]
Lipopeptide biosurfactants	*B. licheniformis* MB01	0.994, 1.008, 1.022, and 1.036 kDa	*Staphylococcus aureus*	[117]
			*Escherichia coli* *Vibrio cholerae* *Vibrio parahaemolyticus* *Vibrio harveyi* *Pseudomonas aeruginosa* *Proteus* species	
Lipopeptide biosurfactants (surfactin homologues and fengycin A, B)	*B. licheniformis* V9T14 (DSM 21038)	ND	*E. coli* CFT073 (biofilm formation)	[118,119]
Surfactin (major isoform—surfactin C)	*B. licheniformis* ATCC 12713	ND	*C. perfringens* ATCC*13124* *Staphylococcus aureus* BCRC10780	[120,121]
			*B. hyodysenteriae* ATCC 27164	
Surfactant BL86	*Bacillus licheniformis* BL86	from 0.979 to 1.091 kDa and varying in increments of 14 Da	ND	[122]
BL1193 (non-lipopeptide type biosurfactant together with lipopeptides, plipastatin, and surfactin)	*B. licheniformis* F2.2	1.193 kDa	*B. subtilis*	[103]
			*Pseudomonas aeruginosa* *Escherichia coli* (inhibited by plipastatin)	
**2.2.2.2. Lichenysins**				
Lichenysin	*B. licheniformis* NBRC 104464	ND		[123]
Lichenysins A	*B. licheniformis* BAS50	1.006–1.034 kDa	*Bacillus* sp. *Strain* ATCC 39307 *Bacillus subtilis* *Staphylococcus aureus*	[124]
			*Acinetobacter calcoaceticus* *Alcaligenes eutrophus* *Pseudomonas fluorescens* *Pseudomonas proteofaciens* *Escherichia coli* *Enterobacter* sp. strain 306	
**2.2.2.3. Biosurfactant licheniformin**				
Licheniformin	*B. licheniformis* MS3	1.438 kDa	ND	[125]
BLS	*B. licheniformis* P40	0.800 kDa	*Bacillus cereus* ATCC 14579 *Bacillus cereus* (food isolate) *Bacillus subtilis* (food isolate) *Corynebacterium fimi* NCTC 7547 *Enterococcus faecalis* (clinical isolate) *Lactobacillus acidophilus* ATCC 4356 *Listeria monocytogenes* ATCC 7644 *Listeria inoccua* (food isolate) *Rhodococcus* sp. *Staphylococcus intermedius* (clinical isolate) *Streptococcus* sp. (b-haemolytic) *Streptococcus* sp. (clinical isolate) *Streptococcus* sp. (clinical isolate)	[29,126]
			*Aeromonas hydrophila* (clinical isolate) *Aeromonas sp*. (clinical isolate) *Enterobacter aerogenes* (food isolate) *Erwinia carotorovora* (food isolate) *Erwinia carotorovora 309* (food isolate) *Erwinia carotorovora A325* (food isolate) *Pasteurella haemolytica* (clinical isolate) *Salmonella gallinarium* (clinical isolate)	
**2.2.2.4. Fengycins**				
Fengycins A,B (and surfactin homologues)	*B. licheniformis V9T14* (*DSM 21038*) (also produced by *Bacillus licheniformis* B6)	ND	*E. coli* CFT073 (biofilm formation)	[118,119,127]
Lipopeptides (fengycins A and B, iturin, kurstakin, surfactin isophorms)	*Bacillus licheniformis* B6	1.2–1.65 kDa	*Staphylococcus aureus*	[127]
			*Escherichia coli* *Klebsiella* sp.	
**2.2.3. Other lipopeptides**				
Biosurfactant	*Bacillus licheniformis* M104	ND	*Bacillus subtilis* ATCC 6633 *Bacillus thuringiensis var. kurstaki* ATCC 19266 *Bacillus thuringiensis* ATCC 10792 *Bacillus cereus* ATCC 9634 *Staphylococcus aureus* ATCC 25928 Methicillin-resistant *Staphylococcus aureus* ATCC 25928	[105]
			*Proteus vulgaris* ATCC 13315 *Pseudomonas aeruginosa* ATCC 10145 *Escherichia coli* ATCC 11775 *Escherichia coli* ATCC 11246 *Salmonella typhimurium* ATCC 14028	
Antiadhesin (I)	*B. licheniformis* 603	ND	*Corynebacterium variabilis* Ac1122	[128]
			*Acinetobacter* sp.	
CB-1	*Bacillus licheniformis*	42 kDa	*Bacillus cereus* NFRI 8004	[104]
			*Escherichia coli* K-12	
NIOT-AMKV06	*Bacillus licheniformis* NIOT-AMKV06	ND	*Enterococcus faecalis* MTCC439 *Bacillus subtilis* MTCC441 *Staphylococcus aureus* MTCC96 *Micrococcus luteus* MTCC1541	[129]
			*Salmonella typhi* MTCC734 *Proteus mirabilis* MTCC142 *Vibrio cholerae* MTCC3904 *Klebsiella pneumoniae* MTCC109 *Shigella flexineri* MTCC1457	

ND—no data. Gram- bacteria are lower the line, and Gram+ bacteria are above the line.

**Table 5 pharmaceutics-15-01893-t005:** Antimicrobial exopolysaccharides produced by *Bacillus licheniformis*.

Substance(s) Specific/Unspecific Name	Producing Strain	Molecular Mass	Activity Assay	Reference
Levan (fructan)	*B. licheniformis* BK1, BK2	~2–100 × 10^3^ kDa	*Staphylococcus aureus*	[188]
			*E. coli* *Pseudomonas aeruginosa*	
EPS1	*B. licheniformis* 24	ND	*Vibrio cholerae* non-O1	[189]
Bl-EPS	*B. licheniformis* Dahb1	ND	*B. subtilis* KT763078.1 *B. pumilus* Dahb3 HQ693273.1	[106]
			*P. aeruginosa* Dahp1 (HQ400663.1) *P. vulgaris* Dahp1 (HQ116441.1)	
EPS-T14	*B. licheniformis* T14	1000 kDa	multiresistant clinical strains: *Staphylococcus aureus*	[190]
			*Escherichia coli* *Klebsiella pneumonia* *Pseudomonas aeruginosa*	
Exopolysaccharide	*B. licheniformis SP1*	1800 kDa	*Escherichia coli* PHL628 *Pseudomonas fluorescens* (only biofilms formation)	[191]

ND—no data. Gram- bacteria are lower the line, and Gram+ bacteria are above the line.

**Table 6 pharmaceutics-15-01893-t006:** Antimycobacterial substances produced by *Bacillus licheniformis*.

Substance Name	Molecular Mass	Sensitivity to Enzymes	Sensitivity to Temperature	Reference
Bacitracin/Ayfivin	1.42 kDa	ND	Resistant to temperature under 60 °C.	[113,130]
Proticin	0.56 kDa	ND	ND	[38]
Peptide A12-C	0.77 kDa	Resistant to trypsin, pronase and proteinase K, carboxypeptidase A, alkaline phosphatase, lipase, lysozyme, β-glucosidase and β-glucuronidase	resistant to heat (100 °C for 30 min at pH 7.0)	[46]
Licheniformins	3.8–4.8 kDa	ND	ND	[37]
Amoebicins d13-A, d13-B, and d13-C	1.87 kDa	Resistant to trypsin, pronase, proteinase, lipase and β-glucuronidase	retained 100% of the activity after being heated at 100 °C for 30 min and also after being stored at −20 °C for 6 months	[108]
Lichenicidin	3 kDa and 3.25 kDa	No loss of antimicrobial activity after treatment with trypsin and chymotrypsin, inactivation by proteinase K and pronase E	supernatant was inactivated within two hours by 80 °C and within 30 min by boiling	[50,52]

ND—no data.

## Data Availability

Data sharing not applicable. No new data were created or analyzed in this study.

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
