# Peer review of "Bacillus licheniformis: A Producer of Antimicrobial Substances, including Antimycobacterials, Which Are Feasible for Medical Applications"

_pharmaceutics, 2023, doi:10.3390/pharmaceutics15071893_

Round 1

Reviewer 1 Report

This paper described or reviewed the antimicrobial substances from Bacillus licheniformis. However, I am concerned with the structure of this paper. 

Please rewrite the abstract section. (abstract should include background, findings, and conclusion)

Please split Table 1 into several tables. This table is too long.

Please added several beautiful Figures.

Please rewrite the conclusion section. right now it is not a formal  conclusion section.

Author Response

We thank the reviewer for valuable suggestions, which helps to improve the visibility of this manuscript. Below is our "point-by-point" response to this reviewer's comments.

Point 1: “This paper described or reviewed the antimicrobial substances from Bacillus licheniformis. However, I am concerned with the structure of this paper. Please rewrite the abstract section. (abstract should include background, findings, and conclusion)”

Response 1:  We have added a phrase in the beginning of the Abstract in order to provide the rationality for this review. However, we should stress that this paper is not an experimental one, which indeed needs to have background, findings etc.  For example, it is not clear the meaning of the results section for review article. For the Abstract writing we have used examples and styles from already published reviews in this Journal.  

Point 2: “Please split Table 1 into several tables. This table is too long.”

Response 2: We have changed it.

Point 3: “Please added several beautiful Figures.”

Response 3:  We have added two figures concerning the classification and mechanisms of action of antibacterial substances produced by Bacillus licheniformis.

Point 4: “Please rewrite the conclusion section. right now, it is not a formal conclusion section.”

Response 4:  We thank the Reviewer for valuable suggestion, we have changed this section accordingly.

Reviewer 2 Report

In this review,Shleeva M.O. et al focused on the current state of knowledge about classes of antibiotic substances produced by B. licheniformis and their properties such as molecular weight, heat stability that may allow a comprehensive perspective of their antimicrobial potential including anti-fungal, amoebolytic  and anti- mycobacterial properties.

This review is comprehensive, but is not clearly written which may be modified as following:

1, Only introduction and other two sections were numbered and “1, Introduction “and” 1 Bacteriocins” made the article confusing. It will be clearer to number all the section and subsection. Same problem is in the table 1.

2, Based on thermostability, size and chemical moieties, bacteriocins are classified into four major groups, according to reference 47. In this review, the authors modified Cotter’s classification for the description of antimicrobial substances: heat-stable and heat-labile proteins larger than 10 kDa were assigned to class III and added class V - proteins with undetermined molecular weight. In the text, there is “1.3. class III – Proteins larger than 10 kDa”(L373) while on the table 1 , there is inconsistent description of “class III – heat-labile proteins larger than 10 kDa” . Heat –stable proteins larger than 10 kDa should be added to table 1.

  3, The authors sub -classified the class II proteins according to their activities, including anti-fungal and amoebolytic activities while did not summarize other substances with such activities, such as bacteriocin which has anti-fungal activity. It is better to describe anti-fungal /amoebolytic active substances to a separate section which including all the reported antimicrobial substances produced by B. licheniformis.

  4, In table 1, it is better to separate G+ bacteria with  G- bacteria, and not to put fungi with bacteria together.

  5, The English language requires editing, since there are many misspells. The article requires careful review. Such as:

In Line 79” which produces grate variety”.

In L116, “When grown on the identical medium, ddifferent strains of”,

L121-123, “When growing on a  medium with lactate and a high ratio of nitrogen and carbon, B. licheniformis Weigmann  emend. Gibson can produce licheniformins,..”

L116-117 and L129-130, “When grown on the identical medium, different strains of B. licheniformis produce 116 a different set of substances with antibacterial activity [31].”Exactly same sentence repeated twice.

B. licheniformis and Aspergillus niger, Penicillium sp.Fusarium sp. Cladosporium sp. should be italic, both in text and in the table.

In Table 1,” o6.4 kDa” should be 6.4 kDa, “3348 kDa” should be 3348 Da”. And the authors should unify when to use kDa and when to use Da. In L439, there is 12, 000 Da. It is better to use 12 kDa.

The English language requires editing, since there are many misspells. The article requires careful review. Such as:

In Line 79” which produces grate variety”.

In L116, “When grown on the identical medium, ddifferent strains of”,

L121-123, “When growing on a  medium with lactate and a high ratio of nitrogen and carbon, B. licheniformis Weigmann  emend. Gibson can produce licheniformins,..”

L116-117 and L129-130, “When grown on the identical medium, different strains of B. licheniformis produce 116 a different set of substances with antibacterial activity [31].”Exactly same sentence repeated twice.

B. licheniformis and Aspergillus niger, Penicillium sp.Fusarium sp. Cladosporium sp. should be italic, both in text and in the table.

In Table 1,” o6.4 kDa” should be 6.4 kDa, “3348 kDa” should be 3348 Da”. And the authors should unify when to use kDa and when to use Da. In L439, there is 12, 000 Da. It is better to use 12 kDa.

Author Response

We thank the reviewer for valuable suggestions, which helps to improve the visibility of this manuscript. Below is our "point-by-point" response to this reviewer's comments.

Point 1:  Only introduction and other two sections were numbered and “1, Introduction “and” 1 Bacteriocins” made the article confusing. It will be clearer to number all the section and subsection. Same problem is in the table 1.”

Response 1:  We have numbered all sections and subsections. Also, we have made corrections to the tables.

Point 2: Based on thermostability, size and chemical moieties, bacteriocins are classified into four major groups, according to reference 47. In this review, the authors modified Cotter’s classification for the description of antimicrobial substances: heat-stable and heat-labile proteins larger than 10 kDa were assigned to class III and added class V - proteins with undetermined molecular weight. In the text, there is “1.3. class III – Proteins larger than 10 kDa”(L373) while on the table 1 , there is inconsistent description of “class III – heat-labile proteins larger than 10 kDa” . Heat –stable proteins larger than 10 kDa should be added to table 1.”

Response 2: We have corrected Table 1.

  Point 3: The authors sub -classified the class II proteins according to their activities, including anti-fungal and amoebolytic activities while did not summarize other substances with such activities, such as bacteriocin which has anti-fungal activity. It is better to describe anti-fungal /amoebolytic active substances to a separate section which including all the reported antimicrobial substances produced by B. licheniformis.”

Response 3: In class II, we describe bacteriocins and classify them according to the object of action. When describing non-ribosomal peptides and exopolysaccharides, we do not classify them according to the objects of action within the groups, but we list these objects for each substance. If we make one more section on substances with antifungal and amoebolytic activity, then the selected main groups of substances will be mixed. But we have made Table 2 (Antifungal substances produced by Bacillus licheniformis) for all antifungals and Table 3 (Amoebolytic substances produced by Bacillus licheniformis) for all amoebolytic substances, including members of different substance groups.

  Point 4: In table 1, it is better to separate G+ bacteria with  G- bacteria, and not to put fungi with bacteria together.”

Response 4: We have moved fungi and amoebas to separate tables. And we separated G+ and G- bacteria in Table 1, 4, and 5.

  Point 5: The English language requires editing, since there are many misspells.

Response 5: We've made the recommended changes and the MS has been thoroughly checked for English via MDPI service.

Point 6: The article requires careful review. Such as:

In Line 79” which produces grate variety”.

Response 6: We have added a correction to the text.

Point 7: In L116, “When grown on the identical medium, ddifferent strains of”,

Response 7:  We have added a correction to the text.

Point 8: L121-123, “When growing on a  medium with lactate and a high ratio of nitrogen and carbon, B. licheniformis Weigmann  emend. Gibson can produce licheniformins,..”

Response 8: Checked and corrected

Point 9: L116-117 and L129-130, “When grown on the identical medium, different strains of B. licheniformis produce 116 a different set of substances with antibacterial activity [31].”Exactly same sentence repeated twice.

Response 9: Corrected

Point 10: “B. licheniformis and Aspergillus niger, Penicillium sp.Fusarium sp. Cladosporium sp. should be italic, both in text and in the table.”

Response 10: Corrected

Point 11: “In Table 1,” o6.4 kDa” should be 6.4 kDa, “3348 kDa” should be 3348 Da”. And the authors should unify when to use kDa and when to use Da. In L439, there is 12, 000 Da. It is better to use 12 kDa”.

Response 11: We have made this correction and choose format “kDa”

Reviewer 3 Report

Review entitled Bacillus licheniformis - a perspective for medical applications producer of variety of antimicrobial substances including antimycobacterials

This is a manuscript with a lot of information about B. licheniformis. I think the paper is well structured, but in places it is hard to follow. The part about the antimycobacterial effect is not very complex, therefore I think this part should be improved by including potential mechanisms of action of selected substances. 

I still have some doubts about some of the information provided, namely - 

How to explain, i.e. what is the mechanism by which (line 129) - strains of B. licheniformis grown in identical medium produce different subsets of substances with antimicrobial potential.  

What is the explanation that bacteriocins have differential antimicrobial potential?

Author Response

We thank the reviewer for valuable suggestions, which helps to improve the visibility of this manuscript. Below is our "point-by-point" response to this reviewer's comments.

Point 1: “This is a manuscript with a lot of information about B. licheniformis. I think the paper is well structured, but in places it is hard to follow. The part about the antimycobacterial effect is not very complex, therefore I think this part should be improved by including potential mechanisms of action of selected substances.” 

Response 1: The mechanism of action of antimycobacterials is known only for three substances: lantibiotics, bacitracin and subtilin. The mechanism of action of lantibiotics and subtilins is described in the section: 2.1.1. Class I: - heat-stable lanthionine-containing peptides smaller than 5 kDa. The mechanism of action of bacitracin is described in the section: 2.2.1. Bacitracin.  For other antimycobacterial substances, the mechanism of action is unknown.

And we added a Figure 2 dedicated to these mechanisms.

Point 2: How to explain, i.e. what is the mechanism by which (line 129) - strains of B. licheniformis grown in identical medium produce different subsets of substances with antimicrobial potential.”  

Response 2: Plausible, all strains of B. licheniformis potentially are able to produce variety of antimicrobial substances, however, the synthesis and production of particular substance can be differently regulated on the transcriptional or translational level for certain strain grown in identical medium. As a result, amount of secreted antimicrobial compounds may substantially vary from strain to strain allowing to consider them as unique producers of defined set of antimicrobials. This remark is added to the text Line 117-123 (before tables).

Point 3: “What is the explanation that bacteriocins have differential antimicrobial potential?”

Response 3: Bacteriocins have a different structure and a different mechanism of action against bacteria. Their ability to reach their targets is crucial for their effectiveness. Due to variations in cell wall structure, certain bacteriocins, particularly those designed to target intracellular components, may encounter challenges in penetrating the cell walls of mycobacteria or Gram-negative bacteria. Conversely, pore-forming bacteriocins have demonstrated a broader spectrum of activity against various types of bacteria. This explanation is added to the text Line 531-536.

Round 2

Reviewer 1 Report

The abstract should mainly show  the findings of this review(results), after you summarized the references. What principle? waht mechanism?..... 

More figures are needed, especially for the latter parts of this review. 

 Too much irrelevant information in the abstract section.

  •  

Author Response

We thank the Reviewer for detailed and useful review of our work. 

We revised our manuscript in appropriate way with reviewers’ comments.

We rewrote the abstract and made it more specific. We have also added four figures regarding the mechanism of action of known antimycobacterial compounds and their structure.